



# Photochemical transformation of residential wood combustion emissions: dependence of organic aerosol composition on OH exposure

Anni Hartikainen[1], Petri Tiitta[1], Mika Ihalainen[1], Pasi Yli-Pirilä[1], Jürgen Orasche[2], Hendryk Czech[1,2,3] Miika Kortelainen[1], Heikki Lamberg[1], Heikki Suhonen[1], Hanna Koponen[1], Liqing Hao[4], Ralf Zimmermann[2,3], Jorma Jokiniemi[1], Jarkko Tissari[1], and Olli Sippula[1,5]

[1] Department of Environmental and Biological Sciences, University of Eastern Finland, Kuopio, FI-70211, Finland
[2] Joint Mass Spectrometry Centre, Comprehensive Molecular Analytics, Helmholtz Zentrum München, Neuherberg, DE-85764, Germany
[3] Joint Mass Spectrometry Centre, Institute of Chemistry, University of Rostock, Rostock, DE-18059, Germany
[4] Department of Applied Physics, University of Eastern Finland, Kuopio, FI-70211, Finland
[5] Department of Chemistry, University of Eastern Finland, Joensuu, FI-80101, Finland

*Correspondence to*: Anni Hartikainen (anni.hartikainen@uef.fi)

**Abstract.** Residential wood combustion (RWC) emits large amounts of gaseous and particulate organic aerosol (OA). In the atmosphere, the emission is transformed via oxidative reactions, which are under daylight conditions driven mainly by hydroxyl radicals (OH). This continuing oxidative aging produces secondary OA and may change the health- and climate-related properties of the emission. In this work, emissions from two modern residential logwood combustion appliances were photochemically aged in an oxidation flow reactor with various OH exposure levels, reaching up to $6 \times 10^{11}$ s cm$^{-3}$ (equivalent to one week in the atmosphere). Gaseous organic compounds were analysed by proton transfer reaction time-of-flight mass spectrometry (PTR-ToF-MS), while particulate OA was analysed online by an aerosol mass spectrometer (AMS) and offline by thermal-optical analysis and thermal desorption-gas chromatography mass spectrometry. Photochemical reactions increased the mass of particulate organic carbon by a factor of 1.3–3.9. The increase in mass took place during the first atmospheric equivalent day of aging, after which the enhancement was independent of the extent of photochemical exposure. However, aging increased the oxidation state of the particulate OA linearly throughout the assessed range, with $\Delta$H:C/$\Delta$O:C slopes between -0.17 and -0.49 in van Krevelen space. Aging led to an increase in acidic fragmentation products in both phases. For the volatile organic compounds measured by PTR-ToF-MS, the formation of small carbonylic compounds combined with the rapid degradation of primary volatile organic compounds such as aromatic compounds led to a continuous increase in both the O:C and H:C ratios. Overall, the share of polycyclic aromatic compounds (PACs) in particles degraded rapidly during aging, although some oxygen-substituted PACs, most notably naphthaldehydic acid, increased, in particular during relatively short exposures. Similarly, the concentrations of particulate nitrophenols rose extensively during the first atmospheric equivalent day. During continuous photochemical aging, the dominant reaction mechanisms shifted from the initial gas phase





functionalisation/condensation to the transformation of the particulate OA by further oxidation reactions and fragmentation. The observed continuous transformation of OA composition throughout a broad range of OH exposures indicates that the entire atmospheric lifetime of the emission, from fresh to shortly aged to long-term aged emission (representative of long-range transported pollutants), needs to be explored to fully assess the potential climate and health effects of OA emissions.

## 1 Introduction

Biomass combustion is a major source of atmospheric particulate matter (PM) and is considered the main anthropogenic source of organic matter and the third largest contributor of black carbon (BC) emissions globally (Klimont et al., 2017). The use of wood fuels in small-scale residential settings is a main source for ambient organic aerosol (OA) in many parts of the world. For example, residential wood combustion (RWC) has been identified as a major source of ambient air fine particles in several European cities, where its relative contribution has been estimated to further increase in the future while PM emissions from other sources, such as industry and traffic, are decreasing (Denier Van Der Gon et al., 2015; Klimont et al., 2017). The amount and contents of the RWC emissions depend greatly on combustion conditions which are generally affected by the combustion procedure, fuel, and appliance technology (Bhattu et al., 2019; Nuutinen et al., 2014; Orasche et al., 2013; Tissari et al., 2009). In logwood-fired appliances, there is also a strong variation in the emissions during the different combustion phases of batches, with ignition producing the highest organic emissions (Bhattu et al., 2019; Kortelainen et al., 2018). However, highest black carbon concentrations are emitted during the flaming phase, while the char burnout phase typically emits large amounts of carbon monoxide but low, mainly inorganic particulate emissions (Kortelainen et al., 2018). Combustion conditions also affect the emissions of many toxic compounds, such as polycyclic aromatic compounds (PACs) (Kim et al., 2013; Orasche et al., 2013), and are consequently strongly linked with the adverse health effects of the emissions (Bølling et al., 2009; Kanashova et al., 2018; Kasurinen et al., 2018).

Many of the main organic species in fresh, unaged wood smoke are connected to the composition of wood fuel, such as levoglucosan, a common biomass burning marker, and lignin degradation products such as methoxyphenols and their derivatives (Elsasser et al., 2013; McDonald et al., 2000; Orasche et al., 2013). In addition, the gaseous organic emission from RWC contains of hundreds of volatile organic species (Bhattu et al., 2019; Bruns et al., 2017; Hartikainen et al., 2018; McDonald et al., 2000). Notably, RWC is an important anthropogenic source for volatile organic compounds (VOCs) with high potential for secondary particulate organic aerosol (SOA) formation. The most potent SOA-precursor compounds include aromatic hydrocarbons and oxygenated species, such as phenolic and furanoic compounds (Bruns et al., 2016; Hartikainen et al., 2018). Furthermore, the gaseous emissions contain high amounts of carbonyls, such as formaldehyde and acetaldehyde, with adverse health effects (Reda et al., 2015; U. S. EPA).

OA has an atmospheric lifetime of approximately one week (Hodzic et al., 2016), during which its chemical composition and potential environmental and health effects are likely to transform extensively. In daylight conditions, hydroxyl radicals (OH) dominate this aging process, where the oxidation of VOCs and semi-VOCs forms products with lower vapour



pressures. These oxidised secondary organic species partition into the particulate phase, resulting in an enhancement of ambient air particulate organic matter concentrations (Robinson et al., 2007). While aromatic VOCs are noted as the main SOA precursors from RWC, the complete pathway for SOA formation and the final SOA yields of complex VOC mixtures under different atmospheric conditions remain unclear (Bruns et al., 2016; Hartikainen et al., 2018; Hatch et al., 2017; McFiggans et

al., 2019). Recent experiments on RWC exhaust estimate the mass of particulate OA to increase by a factor of 1.6–5.3 within approximately one day of photochemical aging (Bertrand et al., 2017; Bruns et al., 2015; Grieshop et al., 2009; Heringa et al., 2011; Tiitta et al., 2016). In addition, the photochemical transformation of the particulate matter may be significant when considering the atmospheric fate of RWC emissions: it has been reported that only a minority of initial biomass burning particulate OA remains unreacted after a few hours of atmospheric aging (Hennigan et al., 2011; Tiitta et al., 2016).

Photochemical aging also decomposes polycyclic aromatic hydrocarbons (PAHs) in both the particulate and the gas phase, which may decrease the carcinogenic properties of the emissions or on the other hand, may lead to the formation of even more toxic, oxygen- or nitrogen-substituted PACs (Keyte et al., 2013; Miersch et al., 2019). For instance, these substituted PACs have been reported to cause a substantial part of the particle-induced mutagenicity in Beijing, with a contribution of only 8 % relative to the concentration of hydrocarbon PACs (Wang et al., 2011). Similarly, oxidation in nitrogen oxide ($NO_x$)-rich

conditions can produce nitrophenols which are harmful to toward plant growth and for human health (Harrison et al., 2005) and have been identified as an important constituent of light-absorbing organic matter ('brown carbon') (Moise et al., 2015; Zhang et al., 2011), thereby affecting atmospheric radiative forcing.

The photochemical aging of RWC emissions has previously been studied in smog chambers (Bertrand et al., 2017; Bruns et al., 2015; Heringa et al., 2011; Tiitta et al., 2016), where the aging was monitored as a batch process from fresh

emission to up to one atmospheric equivalent day of exposure (eqv.d) assuming an ambient average OH concentration of $10^6$ molec cm$^{-3}$; Prinn et al., 2001). As an alternative, oxidation flow reactors (OFRs) with continuous sample flow have been increasingly utilised in combustion emission studies. To achieve similar or higher oxidant exposures than chambers in shorter residence times, OFRs have been used with high ozone concentrations together with high-intensity low-wavelength UV lamps to generate OH-radical concentrations orders of magnitudes higher than those of smog chambers (Kang et al., 2007; Ihalainen

et al., 2019; Simonen et al., 2017). Thus, OFRs enable measurements with better temporal resolution, which is a benefit when assessing the aging of aerosols from dynamic sources, such as batchwise logwood combustion.

In this study, we investigated the photochemical transformation of the OA from two RWC appliances fired with spruce and beech logwood. Atmospheric aging was simulated using the photochemical emission aging flow tube reactor (PEAR) (Ihalainen et al., 2019). In the PEAR, emissions were exposed to a range of OH concentrations, after which the

95 transformation of emissions and related secondary organic emissions were measured with a comprehensive analysis setup (Fig. 1). The OA measurements included gas-phase analysis by proton transfer reactor time-of-flight mass spectrometry (PTR-ToF-MS) and investigation of the particulate phase online by aerosol mass spectrometry (AMS) and offline by targeted gas chromatography mass spectrometry and thermal-optical analyses. With photochemical exposure varying from 0 to 7 eqv.d, we



assessed the transformation of OA composition from fresh emission up to exposures representative of long-range transported smoke.

## 2 Material and methods

### 2.1 Experimental conditions

Experiments were conducted in the ILMARI laboratory of the University of Eastern Finland (www.uef.fi/ilmari) with the experimental setup shown in Figure 1. The two combustion appliances used were modern stoves with improved air intakes. First, a heat-storing masonry heater (Hiisi 4, Tulikivi Ltd., Finland) representing the typical modern logwood combustion technology utilised in Northern Europe, was fired with spruce logwood. The combustion procedure in the masonry heater consisted of three 2.5-kg batches (35-min combustion time) of spruce logwood representing kiln-dried fuel (5 % moisture content, S-5%), after which there was a 25-min char burning period prior to two 45-min batches with moist (22 % moisture content, S-22%) spruce logwood. Second, a non-heat retaining chimney stove (Aduro 9.3, Denmark) representing Middle-European modern logwood stoves was fired with both beech and spruce logs. The chimney stove experiments consisted of five 2-kg batches (combustion time 40–55 min) of beech logwood (17% $H_2O$, B-17%), followed by two 50-min batches of S-22%. Beech was used in these experiments because it is the most common firewood used in Middle Europe, while spruce is used both in Northern and Middle Europe. For ignition, 150 g of dry kindling was placed on the top of the first batch in the cold furnace. Each batch was divided into three parts: ignition, flaming, and burnout phase. The ignition phase was determined to last from the beginning of the batch to the moment of batch maximum flue gas $CO_2$ concentration, and the burnout phase to begin from the moment when the CO concentration started to elevate and remained at a high level until the end of the batch (Fig. 2). The modified combustion efficiency (MCE) as a function of time was calculated from primary flue-gas $CO_2$ and CO concentrations as $\Delta CO_2 / (\Delta CO_2 + \Delta CO)$.

The exhaust was sampled from the stack with a 10 µm pre-cut cyclone. The sample was diluted with a combination of a porous tube diluter and an ejector diluter (Dekati FPS ejector, Finland) and had a dilution ratio (DR) of 40–150 (Table 1) when fed to the PEAR OFR. In the chimney stove experiments, an additional ejector diluter with a DR of 8 (Dekati DI-1000, Finland) was placed before the secondary online aerosol instruments. Compared to the no-aging experiments, a higher DR was selected for the aging experiments in response to the expected increase in particulate matter in the PEAR during photochemical aging. Measured concentrations were corrected for the dilution and normalised to stoichiometric dry flue-gas by multiplication with the stoichiometric correction factor (SCF) of Eq. 1 based on the secondary $CO_2$ concentration and the fact that the dry flue gas of wood combustion with no excess air contains 20.2 % $CO_2$.

$$SCF = \frac{20.2\ \%-CO_{2,background}}{CO_2-CO_{2,background}} \tag{1}$$

In addition, the concentrations were normalised to a 13 % flue-gas oxygen content.



## 2.2 Use of the PEAR

The PEAR (Ihalainen et al., 2019) was used to continuously age the sample stream. In the setup, the extent of photon flux and consequential photochemical aging were controlled by adjusting the voltage of the 254-nm UV lamps. The total flow through the PEAR was 60 L/min. In addition to the 55 L/min sample flow, additional humified and purified air was introduced to the PEAR to obtain a relative humidity of 45 ± 5 %. In the aging experiments, 0.25–0.5 L/min ozone and 15 mL/min of butanol-*d9* were mixed with the main sample flow before the PEAR inlet (Fig. 1). In the PEAR, the photolysis of the ozone formed

hydroxyl radicals via reactions (2) and (3). Thus, the extent of photochemical aging in the PEAR depended on the photon flux inside the reactor and the introduction of OH precursors, namely, the externally fed $O_3$ and $H_2O$.

$$O_3 + hv \rightarrow O_2 + O(^1D) \tag{2}$$

$$O(^1D) + H_2O \rightarrow 2\ OH \tag{3}$$

The OH exposure ($OH_{exp}$) was determined continuously according to the method presented by Barmet et al. (2012) by

introducing a constant flow of butanol-*d9* to the PEAR. The $OH_{exp}$ for clean air in the PEAR prior to the exhaust input ranged from $8.2 \times 10^{10}$ to $1.6 \times 10^{12}$ molec $cm^{-3}$ s, which is equivalent to 1–18 days in an ambient atmosphere (eqv.d) with an estimated OH concentration of $1 \times 10^6$ molec $cm^{-3}$ (Prinn et al., 2005). The alternate reaction pathways in the PEAR, namely, exposure to photolysis ($flux_{254nm,exp}$), excited ($O(^1D)_{exp}$), atomic oxygen ($O(^3P)_{exp}$), and ozone ($O_{3,exp}$), were assessed using the OFR exposure estimation equations of Peng et al. (2016) for OFRs with 254-nm lamps. Both the $OH_{exp}$ and the alternate reaction

pathways were affected by the external OH reactivity ($OHR_{ext}$) (Li et al., 2015) in the PEAR. The $OHR_{ext}$ was calculated with Eq. (4) from the concentrations ($c_i$) of the primary gaseous compounds in the PEAR, with their OH reaction rate constants $k_i$ (Table S1).

$$OHR_{ext} = \sum k_i c_i \tag{4}$$

Particulate wall losses inside the PEAR were minimised by conductive stainless-steel walls, laminar flow, and a relatively

small surface-to-volume ratio (2.28 $m^2$:139 L) (Ihalainen et al., 2019). The particulate losses were estimated by the loss of elemental carbon (EC), determined from the difference between the EC concentrations measured upstream and downstream of the PEAR, whereas losses of the low-volatility organic compounds (LVOC) were modelled based on the model of Palm et al. (2016). See Section S2.2 for further information on LVOC fate estimation.

## 2.3 Offline filter sampling and analysis

$PM_1$ filter samples were collected on Teflon (PTFE, Pall Corporation, P/N R2PJO47) and quartz fibre  (QF, Pall Corporation, Tissuquartz) filters simultaneously from primary and secondary exhaust at a 10-L/min flow rate, following the methodology presented by Sippula et al. (2009). A pre-impactor (Dekati PM-10 impactor) was used to separate the particles with aerodynamic diameters less than 1 µm ($PM_1$). For the masonry heater, samples were collected separately from the S-5% combustion (100-min collection time) and S-22% combustion (90-min collection time). From the chimney stove, two sample

pairs were collected from the combustion of beech: full first two batches (85-min collection time), and fourth and fifth batches





excluding the last 15 minutes (85-min collection time). The third chimney stove collection consisted of the combustion of two full batches of moist spruce (100-min collection time).

The Teflon filters were weighted before and after sample collection to determine the total $PM_1$ mass of the emission. The amount of organic (OC) and elemental carbon (EC) in $PM_1$ was determined from the QF filters by using a thermal-optical
carbon analyser (Sunset Laboratory Inc.) following the protocol NIOSH5040 (NIOSH, 1999). In-situ derivatisation thermal desorption–gas chromatography–time-of-flight mass spectrometry (IDTD-GC-ToFMS) (Orasche et al., 2011), was applied to the analysis of semi-VOCs in the particulate phase from the QF filters. Non-polar and polar compounds were identified and quantified using mixtures of isotope labelled internal standards and calibration standards; see Supplementary Information Chapter S6 for further information of the analysis procedure.

**2.4 Online aerosol measurements**

Fourier transform infrared spectrometer (FTIR DX4000, Gasmet Technologies Inc.) was implemented on the stack to measure the amount of $NO_x$, $CO_2$, CO, and 27 VOCs (Table S1) in the fresh exhaust. The measured primary VOCs were grouped into four subgroups: alkanes, oxygenated compounds, and unsaturated and aromatic hydrocarbons. After the PEAR, the concentrations of $CO_2$, $O_3$, $NO_x$, and $SO_2$ were monitored with a trace-level single-gas analyzers (ABB $CO_2$ analyser, Siemens),
and organic gaseous compounds in the mass-to-charge (m/z) range of 40–180 by PTR-ToF-MS (PTR-TOF 8000, Ionicon Analytik, Innsbruck, Austria), with $H_3O^+$ as the reagent ion and an electric field to gas number density ratio (E/N) of 130. Mass calibration was done with $H_3O^+$ (m/z 21.02) and 1,3-diiodobenzene (m/z 203.94), which was added as a calibrant for higher m/zs. The processing of the PTR-ToF-MS data was done in a manner similar to that of earlier work (Hartikainen et al., 2018). Reaction rates by Cappellin et al. (2012) were used when available; for other compounds, the reaction rate with $H_3O^+$ was
assumed to be $2 \times 10^9$ $cm^3$ $s^{-1}$ (Table S6).

Particle concentrations and mobility size distributions were monitored with scanning mobility particle sizers before (SMPS 3082, TSI, size range 14.6–661.2 nm) and after PEAR (SMPS 3080, TSI, size range 15.1–661.2 nm). An electrical low pressure impactor (ELPI 10 L/min, Dekati) measured the particle aerodynamic size distribution and the concentration of primary particles in the size range 18.6–5950 nm. The total particulate mass after PEAR was measured with a tapered element
oscillating microbalance monitor (TEOM, Model 1405, Thermo Scientific).

The composition of submicron particulate matter after PEAR was measured by soot particle aerosol mass spectrometer (SP-HR-ToF-AMS, Aerodyne Research Inc). The AMS data was analysed using the standard analysis tools SQUIRREL v1.62A and PIKA v1.22D adapted in Igor Pro 8 (Wavemetrics). The elemental analysis of the OA was conducted using the Improved-Ambient method (Canagaratna et al., 2015), and the average carbon oxidation state ($OS_C$) of the OA was
estimated as $OS_C \approx 2 \times O{:}C - H{:}C$ (Kroll et al., 2011). The AMS OA mass spectra was further examined by positive matrix factorisation (PMF) using the method described in Lanz et al. (2007) and Ulbrich et al. (2009). The PMF Evaluation Tool v.3.05 was applied, and the standard data pre-treatment process was completed based on Ulbrich et al. (2009), including the application of minimum error criteria and down-weighting weak variables as well as m/z 44 ($CO_2^+$) and water-related peaks.





The final four-factor PMF solution covered 98 % of the OA spectra (2.2 % residual). Additional factors did not increase the realistic physical meaning of the solution, while fewer factors were insufficient for a meaningful presentation of the data. The factor identification was confirmed by comparing the time series and mass spectra of each factor with external tracers (nitrate, sulphate, ammonium, chloride, PAH, $CO_2^+$, $C_2H_3O^+$, $C_4H_9^+$, and $C_2H_4O_2^+$). Furthermore, the factors were compared to logwood combustion mass spectra measured by Tiitta et al. (2016) from the aging of spruce logwood exhaust in a smog chamber. The agreement of spectra was denoted with both a coefficient of determination ($R^2$) and the angle between two mass spectra vectors (Kostenidou et al., 2009), where an angle less than 15° indicates a good agreement between two mass spectra.

The polycyclic aromatic hydrocarbons (PAH) and other polycyclic aromatic compounds (PAC) in the exhaust were analysed by using the PAC molecular ions as a proxy, following the P-MIP methodology presented by Herring et al. (2015). The base molecular ions $[M]^+$, their fragments ($[M-H]^+$ and $[M-2H]^+$) and isotopes ($^2H$, $^{13}C$, $^{13}C_2$, $^{15}N$, $^{17}O$, and $^{18}O$) were isolated and quantified using the AMS high-resolution analysis software tool (version 1.22D). The targeted ions included those previously connected with PACs (Herring et al., 2015) and compounds typically released from residential wood combustion (Avagyan et al., 2016; Bertrand et al., 2018; Bruns et al., 2015; Czech et al., 2018; Miersch et al., 2019). The 61 PACs considered (Table S8) were separated into five subgroups: unsubstituted PAHs (UnSubPAHs), oxygenated PAHs (OPAHs), methylated PAHs (MPAH), nitrogen-substituted PAHs (NPAHs), and amino PAHs (APAHs). See Supplementary Information Chapter S5 for further information on AMS analyses.

## 3 Results and conclusions

### 3.1 Combustion conditions

The average modified combustion efficiency was greater than 0.97 for the masonry heater and greater than 0.95 for the chimney stove, with lower MCE occurring mainly during the char burnout periods (Fig. S1). These values are typical for modern batch-wise operated logwood appliances (Bhattu et al., 2019; Czech et al., 2018; Heringa et al., 2011). In addition to the variation within a combustion batch, there were also differences between the individual batches (Fig. 2). Notably, the combustion conditions during the first batch were distinct from later batches. This was a result of the lower initial temperature which caused a longer ignition period (determined by the rising $CO_2$ concentrations), which lasted for 24 ± 5 % (masonry heater) and 35 ± 4 % (chimney stove) of the total duration of the batch; whereas in the later batches, the higher initial temperature shortened the ignition to 9 ± 3 % (masonry heater) or 8 ± 4 % (chimney stove) of the total combustion time. Furthermore, the flaming phases of the first batches were shorter than those of the following batches of dry spruce or beech fuels. The emission profiles were affected by these batchwise differences, with ignition being the period for enhanced organic emissions, whereas the flaming phase was characterized by an increase in particulate emissions consisting mainly of black carbon, as expected based on previous work (Kortelainen et al., 2018). The char burnout phases with these fuels were characterised by high CO concentrations, whereas in moist spruce combustion elevated CO concentrations were measured throughout the batch, thus making the burnout phases less distinguishable.



### 3.2 Primary emissions

#### 3.2.1 Gaseous organic emissions

The primary organic gaseous emissions were measured constantly by an FTIR multicomponent analyser from the undiluted flue gas. Additionally, the PTR-ToF-MS measured diluted, unaged emissions during the no-aging experiments. These datasets

complemented each other because, while FTIR measured only 27 typical combustion-derived VOCs (Table S1), a more detailed insight of the VOC composition was acquired via the PTR-ToF-MS detection of 127 different molecular formulas for gaseous species in the primary aerosol (Table S6). However, PTR-ToF-MS is unable to detect for example alkanes because of their low protonation efficiency. Furthermore, the fragmentation of the PTR-ToF-MS limited in the quantification of compounds with similar mass-to-charge ratios as those of common fragment ions, including unsaturated aliphatic compounds

such as propene ($C_3H_6^+$) at m/z 41.04 or butene ($C_4H_8^+$) at m/z 57.08; however, these compounds were detected using the FTIR.

The major VOC groups measured by PTR-ToF-MS were carbonyls, aromatic hydrocarbons, furans, and phenols (Fig. 6). In addition, unsaturated aliphatic compounds constituted a substantial fraction of VOC, as measured using the FTIR (Fig. 3). The total VOC emissions based on FTIR were $42.9 \pm 10.5$ and $102 \pm 28.0$ mgC m$^{-3}$ for dry and moist spruce combustion in the masonry heater, respectively, and $89.0 \pm 13.5$ and $147 \pm 24.9$ mgC m$^{-3}$ for beech and moist spruce in the chimney stove,

respectively. Thus, the lowest VOC concentrations were measured from the dry spruce combustion. Moist spruce combustion produced a factor of 1.7–3.6 higher emission than dry wood, which is well in line with studies by e.g. McDonald et al. (2000), where moist fuel produced 2–4 times more VOC than dry fuel combustion. The difference between concentrations was highest for oxygenated compounds (factor of $4.2\pm1.7$) and for unsaturated compounds ($2.7\pm0.6$), but significant for all subgroups (paired t-test p-values $\leq0.02$ for all groups for the consecutive dry/moist experiments; Table S2). Differences in VOC emissions

were also observed between the masonry heater and the chimney stove. Emissions from moist spruce combusted in the chimney stove were higher by a factor of 1.5 (p-value 0.05) compared to the masonry heater, with a significant increase in unsaturated and aromatic hydrocarbons (factors of 1.6 and 1.7, respectively, p-values <0.01) based on FTIR measurements.

The VOC emissions of the first batches exhibited a distinct time-dependent behaviour in comparison to the following batches; i.e., the emitted concentrations always increased both after ignition and at the end of flaming phase (Fig.

2), whereas in the following batches there was a sharp emission peak at ignition, after which the concentrations declined as soon as the flaming phase began. These findings agree with those by Kortelainen et al. (2018) and are influenced by the fact that cold ignition is performed from the top of a fuel batch, while the following batches are ignited from the bottom by the glowing embers. VOC emissions were lowest during the burnout period when most of the fuel was already consumed (Fig. S2). Thus, VOC emissions have a reverse time profile to the CO emissions, which peak during the char burnout phase.

Furthermore, the contribution of the different organic compound groups to the total VOC concentration differed in relation to time (Fig. S3). For example, the importance of aromatic species increased for dry fuels (S-5% and B-17%) for the flaming and burnout phases, while other species were pronouncedly emitted at ignition. In addition, the composition of aromatic species measured with PTR-ToF-MS in the unaged emission depended on the phase (Fig. S4). In the masonry heater, aromatic



hydrocarbons (ArHC) had a highest contribution during the ignition phase, but their relative share decreased during flaming

phase, while the share of furanoic and phenolic compounds increased. The relative importance of furanoic and phenolic

compounds in the fresh exhaust of the flaming phase from a masonry heater has been also previously established (Czech et al.

2016). Overall, the share of ArHC in the fresh exhaust is higher and less phase-dependent for beech combustion in chimney

stove than for other experiments. The share of N-containing aromatic compounds, namely, nitrophenol and -cresol, also

increased after ignition. These findings are important also when considering the potential of SOA formation, as aromatic VOCs

have been observed to be the major SOA precursors in RWC emissions (Bruns et al., 2016; Hartikainen et al., 2018).

### 3.2.2 Particulate emissions

The average primary $PM_1$ mass emissions ranging from 33 to 67 mg m$^{-3}$ and the number emissions from $3.2 \times 10^7$ cm$^{-3}$ to 5.4

$\times 10^7$ cm$^{-3}$ (Table 2) were on a similar level with earlier studies reporting emissions from modern logwood stoves (Kortelainen

et al., 2018; Nuutinen et al., 2014; Tissari et al., 2009). The combustion of dry spruce in a masonry heater emitted 1.5- and 2

-fold $PM_1$ mass compared to that of moist spruce and beech, respectively, mainly because of the higher elemental carbon

emissions. Moist spruce generated similar $PM_1$ emissions with both combustion appliances. The organic carbon to elemental

carbon ratio (OC:EC) of dry spruce combustion was very low ($0.07 \pm 0.02$, Table 2) compared to that of moist spruce

combustion ($0.31 \pm 0.45$ for masonry heater and $0.25 \pm 0.04$ for chimney stove) and beech combustion ($0.15 \pm 0.04$). Such

low OC:EC ratios have been previously reported for emissions from modern masonry heaters operating with dry logwood

(Czech et al., 2018; Miersch et al., 2019; Nuutinen et al., 2014).

The particle size distribution (Figs. 4 and S5) from dry spruce combustion was clearly distinguishable from that of

other wood fuels and showed considerably larger mean particle mobility size (GMD 95.5 nm) compared to those of other fuels

(GMD 52.8–68.4 nm). The soot-dominated composition of S-5% exhaust likely increases the GMD, because soot particles are

typically present as larger agglomerates than particles of inorganic origin (ash) which mainly form ultrafine particles (Tissari

et al., 2008). The size distribution and number concentration of particles in an exhaust are not only important because of their

link to potential health effects, but also during aging of the exhaust, because they affect the condensation sink (CS) (Lehtinen

et al., 2003) of condensable vapours during the dilution and aging process. Thus, the particle number concentrations and size

distributions affect the fate of condensable vapours and the overall OA enhancement ratios. The fates of low-volatility

condensable vapours are further discussed in the chapter 3.3.1.

The thermal-optically measured elemental carbon in both the primary and the secondary exhausts correlated well with

the refractory black carbon (rBC) measured by AMS from the secondary exhaust ($R^2 = 0.74$ and 0.76, respectively; see Fig.

S14). Analogous to the elemental carbon results, rBC emission was highest during the combustion of kiln-dried spruce (49.3

$\pm 13.7$ mg m$^{-3}$). Considering the different combustion phases, rBC emissions were highest during high-temperature, flaming

combustion as previously noted also by e.g. Kortelainen et al. (2018). For moist spruce, the rBC concentration dropped to 24.8

$\pm 12.3$ mg m$^{-3}$, likely because of the lower temperature and consequentially slower burn rate. Similar rBC concentrations (24.0

$\pm 6.0$ mg m$^{-3}$) were also measured from spruce combustion in the chimney stove. Unlike the masonry heater where the rBC



concentrations were similar throughout the three dry batches, the rBC concentration from the combustion of beech in a chimney stove decreased considerably after the first batch, from $57.7 \pm 7.2$ mg m$^{-3}$ to $15.6 \pm 5.2$ mg m$^{-3}$. While aging had no effect on the rBC mass, it plays an important role in the formation of SOA by acting as a seed for condensation during aging.

Furthermore, soot cores composed of elemental carbon are chemically active and may enhance the photooxidation of an OA condensed onto soot agglomerates through electron transfer (Li et al., 2018).

### 3.3 OFR conditions

### 3.3.1 Wall losses and fate of condensable vapours

The loss of primary particles in the PEAR was estimated at 6 % based on the thermal–optical measurements of elemental

carbon. When estimating the fate of the LVOC in the PEAR, three possible depletion pathways were considered: condensation onto particles, reactions with OH radicals, and condensation onto walls (Palm et al., 2016). To estimate the fraction of LVOC condensing onto particle phase, condensation sinks (Fig. 4) were calculated according to Lehtinen et al. (2003) using an average of the particle size distribution before and after PEAR. Similar to earlier studies, the downstream particle number concentration and condensation sinks were influenced by new particle formation (Fig. S13) with the extent of nucleation depending strongly

on the aging conditions (Bruns et al., 2015; Ihalainen et al., 2019; Simonen et al., 2017). Long aging conditions greater than 4 eqv.d led to particularly strong increase in the amount of sub-50 nm particles. The new particle formation occurred mainly during the periods when the concentrations of condensable vapours were high; that is, most notably during the ignition phase. However, the distinct formation of ultrafine particles does not necessarily represent the atmospheric fate of condensable vapours because of the faster-than-ambient oxidation resulting in faster gas-to-particle conversion and higher saturation ratios

in the PEAR. At lower OH exposures, the condensation growth was mainly observed as an increase in the number concentration in the larger size range, while less nucleation to new particles was observed than that in long aging experiments.

The importance of different LVOC fates is shown in Figure 5. The majority of the LVOCs condensed onto the particles, and a portion also depleted by reactions with oxidants in the PEAR, while the LVOC wall losses were primarily below 2 %. The amount of LVOCs estimated to exit the PEAR as gas-phase LVOCs exceeded 0.1 % only during the middle

burnout phase of the masonry heater combustion because of the low particle number concentrations during the char burning phase between the change from dry spruce to moist spruce combustion, which generated a low condensation sink into the PEAR.

### 3.3.2 Photochemical aging conditions

During photochemical aging, external OH reactivity (OHR$_{ext}$) is an important parameter affecting OH-radical consumption

and reaction pathways of organic species. For the masonry heater experiments, the average OHR$_{ext}$ was 98–107 s$^{-1}$ for dry spruce samples and 125–172 s$^{-1}$ for moist spruce. In the chimney stove, the average OHR$_{ext}$ was 237–252 s$^{-1}$ for beech and 278–308 s$^{-1}$ for moist spruce samples, except for the low-DR experiment (Exp. 5, DR of 30) where the average OHR$_{ext}$ reached



and 1110 s$^{-1}$ for the beech and moist spruce samples, respectively. The OFR conditions were divided into good, risky, and bad based on the ratio of the photon flux exposure to OH$_{exp}$ by the definitions of Peng and Jimenez (2017) (Fig. S7). Here, conditions were defined as mainly "risky" ($4 \times 10^5$ cm s$^{-1}$ < flux$_{254nm,exp}$/OH$_{exp}$ < $10^7$ cm s$^{-1}$) during all the experiments. The OHR$_{ext}$ varied considerably during combustion cycles, but the limit for bad conditions was exceeded briefly (up to 7 % of total experiment time) only in a few experiments during ignition when the OHR$_{ext}$ peaks above 1000 s$^{-1}$. In terms of OA emission, the limit for bad conditions was exceeded in one masonry heater experiment (S-22%, Exp. 5), where 4 % of OA mass was emitted during this period. For the chimney stove experiments, bad conditions accounted for 2–9 % of OA emissions, excluding the S-22% combustion for the low-DR experiment (46 % of OA emitted under 'bad' conditions).

The OHR$_{ext}$ calculations were limited to the compounds measured from the primary exhaust with FTIR, with NO, CO, and unsaturated hydrocarbons as the main OHR$_{ext}$ producers (Fig. S6). In other words, the products of later-generation oxidation or from fragmentation from the particulate phase were not considered. In addition, the prevailing NO$_x$ concentrations input to the PEAR ranged on average from 150 to 420 ppb, except for the low-DR experiment at a concentration of 1140 and 751 ppb for beech and spruce combustion, respectively. NO is rapidly oxidized to NO$_2$ with the addition of O$_3$ and then partially to particulate nitrate, and the subsequent low-NO conditions in the PEAR reduced the reactions of organic peroxy radicals (RO$_2$) with NO.

The average batchwise OH exposure in the PEAR during photochemical aging experiments ranged from 0.5 to 7 eqv.d depending on the applied photon flux and the OHR$_{ext}$ of the sample. During the experiments, the photochemical exposure varied in line with the varying OHR$_{ext}$ during batchwise combustion, with the highest exposure occurring during the flaming phase and lower exposure during the ignition period (Fig. S8). The extent of alternative non-OH reaction pathways during photochemical aging in the PEAR were compared to the tropospheric conditions (Chapter S2.3), and the exposures to O($^1$D) and O($^3$P) in the PEAR were shown to be similar to those in ambient conditions (Fig. S9), excluding the ignition period during the low-DR experiment, where the importance of O($^3$P)$_{exp}$ briefly exceeded ambient conditions. However, the exposure to ozone in relation to OH radicals was estimated to be lower in the PEAR than in the troposphere (O$_{3,exp}$/OH$_{exp}$<$10^5$), thus leading to more OH-dominated chemical reactions when only considering the initial O$_3$ input. However, our estimations were based on the initial O$_3$ concentrations, whereas O$_3$ is expected to form in the PEAR during photochemical aging because it is a product of the OH + VOC reactions (Carter, 1994). This may have led to the O$_{3,exp}$/OH$_{exp}$ being slightly underestimated.

Several of the emitted VOCs were susceptible to photolysis at 254 nm, most importantly the aromatic hydrocarbons, which are among the main SOA precursors from RWC. Photolysis as a degradation pathway can exceed OH reactions also in the atmosphere for compounds with high photolysis rates (Hodzic et al., 2015). Here, the importance of photolysis was notable for e.g. benzene, of which more than 40 % may have degraded via photolysis (Fig. S10). However, the importance of photolysis during photochemical aging is inversely proportional to the ratio of the OH reaction rate to the photoabsorption cross section ($\sigma_{abs}$), and it can be considered a minor pathway for other main VOC species (Table S5), including other aromatic species such as toluene (<10 % of total degradation during these experiments, see Fig. S10).



### 3.4 Transformation of gaseous phase

The composition of the VOCs changed throughout the studied exposure range, and the abundance of the secondary VOCs increased with an increase in photochemical exposure. However, while the small oxidised organics became increasingly dominant in the gas phase, photochemical aging particularly decreased the share of aromatic compounds (Fig. 6). Namely, the

VOCs measured with PTR-ToF-MS from the aged gas phase were governed by small carbonyls and fragmentation products such as acetic acid ($C_2H_4O_2$, m/z 61.02). The evolution of the total chemical composition is visualised in the van Krevelen diagram, which is often used for simplified characterisation of particulate OA (Heald et al., 2010) and was here extended for the investigation of the gaseous organic phase measured by PTR-ToF-MS (Fig. 7). Photochemical aging caused several simultaneous and subsequent functionalisation reactions of the organic compounds, and the increase in the average H:C ratio

together with an increasing O:C ratio led to linear slopes from +0.69 to +1.0 in the van Krevelen diagram. Most of the VOCs in primary RWC exhaust have relatively high OH reactivities (Fig. S11), and thus their reactions are expected to take place within the first atmosphere equivalent hours of photochemical aging (Bruns et al., 2017; Hartikainen et al., 2018). This indicates the continuous contribution from the secondary products, including the oxidised fragmentation products from the particulate phase. In addition, the loss of compounds with low H:C and O:C ratios, such as aromatic species, is important for

the change of the average composition. Furthermore, the oxidation of compounds undetected by the PTR-ToF-MS, such as alkanes, may introduce secondary products with higher proton affinities and their consequent appearance in the spectra. Conversely, aging may also lead to the growth of compounds outside the considered mass range (m/z 40–180). Compared to previously measured changes of RWC exhaust in a chamber (Hartikainen et al., 2018), aging in the PEAR led to a slightly higher increase in the H:C ratio. This difference implies an increased fragmentation which is likely the result of faster aging

process in the PEAR compared to that in a smog chamber.

Aromatic compounds consisting of aromatic hydrocarbons (ArHCs), phenols, furans, N-containing aromatic compounds (N-aromatics), and other oxygenated aromatics were important constituents of the primary organic gas phase and formed 37–39 % of the fresh VOCs from the masonry heater and 33–34 % from the chimney stove (Fig. 9). Similar shares have been measured from fresh RWC exhaust by Bruns et al. (2017) (13–33 %) and Hartikainen et al. (2018) (33 –36 %).

However, after 2 eqv.d of exposure, their share decreased to less than 20 % of the identified VOCs, which agrees with previously reported conversions of aromatics during photochemical exposure (Bruns et al., 2017; Hartikainen et al., 2018). Overall, the photochemical reactions of aromatics are an important source of SOA because they form products that efficiently partition into the particulate phase. However, there are large differences between the conversion efficiencies of aromatic compound groups. While ArHCs comprise approximately half of the aromatic VOCs in fresh exhaust, their share grows to

over 70 % in 5–7 eqv.d, while the share of oxygenated aromatics decreases with aging in line with their higher OH reactivity. Similar aromatic behaviour in RWC exhaust was observed earlier in a chamber with spruce exhaust (Hartikainen et al., 2018), where the molar share of ArHCs in the total aromatic content increased from 45 % and 32 % to 63 % and 54 % during aging of 0.6 and 0.8 eqv.d, respectively, while the share of furanoic and phenolic species decreased. Furthermore, N-aromatics were



not detectable here with ages exceeding 1 eqv.d, although they have been observed to form with shorter exposures (Hartikainen
et al., 2018). The N-aromatics produced by the first stages of aging may have partitioned to the particulate phase but are also
degraded by subsequent reactions with OH. Simultaneously, the share of aliphatic nitrogen compounds (CHN and CHNO) to
the total concentration remained relatively stable.

Carbonyls were divided into primary and secondary subgroups based on their behaviour during aging (Table S6). The
primary carbonyl group, consisting mainly of acetaldehyde and to a smaller extent of compounds such as acrolein and
butadiene, was prevalent in the fresh exhaust, but their share of identified compounds decreased from 13–27 % in unaged
exhaust to 3–12 % in the highest exposures. This is the result of both the degradation of these compounds and the introduction
of high amounts of carbonyls in the secondary carbonyl group. The ratio between the two carbonyl groups increased linearly
with age (Fig. S12). The secondary carbonyl group was dominated by acetic acid, which was the most prevalent compound
after extensive aging in all experiments and covered over 30 % of the total measured VOC concentration from the highly aged
S-5% and B-17% exhaust. The mainly small acidic compounds in the secondary carbonyl group were formed from the
photochemical reactions of VOCs and from particulate OA, which is a consistent source of oxygenated VOCs such as acetic
acid, formic acid, acetaldehyde, and acetone (Malecha and Nizkorodov, 2016). Of these, acetaldehyde is classified as a primary
carbonyl because it reacts with OH two orders of magnitude faster than the others (Atkinson et al., 2001), and thus its
concentration remains stable throughout the aging process.

**3.5 Transformation of particulate phase**

**3.5.1 OA enhancement and composition**

The photochemical aging process increased the mass of particulate organic carbon measured with thermal–optical carbon
analysis by factors of 1.3 and 3.9 for dry and moist spruce combustion in a masonry heater, respectively, and by factors of 2.0
and 3.0 for beech and moist spruce combustion in a chimney stove, respectively. This agrees well with the previously observed
AMS-based OA enhancement factors (1.6–5.3) for RWC (Bertrand et al., 2017; Bruns et al., 2015; Grieshop et al., 2009;
Heringa et al., 2011; Tiitta et al., 2016) and with the thermal-optical analysis-based organic carbon enhancements previously
measured for the same chimney stove (1.3–1.4 after 1.7–2.5 eqv.d) (Miersch et al., 2019). The organic carbon concentrations
after the PEAR correlated well with the OA measured with AMS ($R^2$ = 0.85, Fig. S14). Aging led to a linear increase in the
AMS-derived ratio of organic matter to organic carbon (OM:OC, Fig. S16), which rose from the initial average ratio of 1.8–
2.2 to 2.7–3.0 during extended aging. Similarly, the $OS_C$ of the OA increased as a function of the photochemical age throughout
the tested exposure ranges (Fig. S16), indicating the existence of continuing reactions of the particulate phase after the rapid
consumption of the majority of the primary gaseous SOA precursors with relatively high OH reaction rates (Fig. S11). In
contrast, as expected, the organic carbon mass enhancement did not increase with continuous aging because the major SOA
precursors were already consumed by relatively short OH exposures. Continuing photochemical exposure may instead reduce
the amount of particulate organic carbon (Kroll et al., 2015), which acts as a source for acidic compounds during photochemical





aging (Malecha and Nizkorodov, 2016; Paulot et al., 2011). See Table S7 for experiment-wise average concentrations and oxidation states.

The oxidation states measured after PEAR without oxidative aging were highest for the chimney stove OA, with average unaged $OS_{CS}$ of 0.22 and 0.41 for beech and spruce combustion, respectively, while dry and moist spruce in the masonry heater had average unaged $OS_{CS}$ of -0.18 and 0.15, respectively. This indicates the existence of different combustion conditions in the studied appliances, with the masonry heater having lower emissions of highly oxygenated compounds and a higher share of unsaturated hydrocarbons compared to that of the chimney stove emissions. However, as a result of oxidative aging, $OS_C$ surpassed 1.5 after 5 eqv.d regardless of the type of experiment. A higher initial oxidation state in the chimney stove exhaust led to shallower slopes (-0.17 and -0.34) in the van Krevelen diagram than those in the masonry heater exhaust (-0.46 and -0.49, Fig. 8). Previously, the aging of spruce combustion exhaust from a masonry heater in a chamber has produced similar but slightly steeper van Krevelen slopes of -0.64 – -0.67 (Tiitta et al., 2016). The steepness may be affected by the lower aging and consequent lower final $OS_C$ (maximum of +0.14) as the O:C ratio has been noted to level off at higher oxidation states (Ng et al., 2011). Slopes are also positively affected by fragmentation which may be enhanced in the PEAR because of the more intensive UV radiation than in smog chambers or in ambient conditions. The $OS_C$ of highly aged exhaust exceeds that observed in ambient aerosol (Kroll et al. 2011, Ng et al. 2011), likely because highly aged aerosol in ambient air is mixed continuously with fresh and less oxidised sources. However, the chemical evolution of OA in the PEAR did follow a similar trend to that observed for semi-volatile oxygenated aerosol which had a van Krevelen slope of -0.5 pointing to e.g. simultaneous fragmentation and acid-group addition (Ng et al., 2011). Furthermore, the changes in $OS_{CS}$ were similar to those previously observed for ambient aerosol aged extensively in an OFR ($OS_C$ up to 2, Ortega et al., 2016), and the van Krevelen slopes agreed well with those previously measured from single precursors in an oxidation reactor, such as -0.48 for toluene or -0.46 for xylene (Lambe et al., 2011).

The evolution of particulate organic aerosol was assessed also by the IDTD-GC-ToFMS analysis of filter samples. When comparing the concentrations in the exhaust after PEAR, the concentrations of compounds with high oxidation states and low number of carbons ($n_C$) increased during photochemical aging (upper-right corner of Fig. 10). The locations of the measured organic compounds in the $OS_C:n_C$ space are shown in Figure S19, and their dilution-corrected concentrations downstream the PEAR in Table S12. In the $OS_C:n_C$ space, the compounds which exhibited a major increase during photochemical aging were located in or above the location of the low-volatility oxidised organic aerosol (LV-OOA) classified by Kroll et al. (2011). These compounds are products of the multistep oxidation process including both fragmentation and oxidative reactions.

OA formation is tied to the availability of organic precursors, and thus the formation of SOA was highest at the ignition phase of each batch (Fig. S15). In the masonry heater, the aged particulate OA mass increased considerably with the introduction of moist logs, simultaneously with the increase in gaseous organics in the fresh emissions. In contrast, the low organic emission by dry spruce combustion was reflected as a lower SOA formation. Another aspect related to the primary exhaust was the extent of OH exposure in the PEAR, which was directly connected to the sample concentrations, as discussed





in Chapter 3.3.2; namely, the photochemical aging was lower during periods of high emission, leading to lower oxidation
states for the OA emitted during ignition.

### 3.5.2 PMF analysis of particulate OA composition

PMF analysis applied to the exhaust produced a four-factor solution for the OA covering 98 % of the spectra. The spectra of
the factors are shown in Figure S17. Two of the factors were associated in particular with the primary OA from biomass
combustion: pyrolysis-BBOA, formed especially during ignition, and flaming-BBOA, emitted pronouncedly during the
flaming phase. The other two factors, semi-volatile oxygenated OA (SV-OOA) and low-volatility oxygenated OA (LV-OOA),
represent oxygenated organics with notably higher $OS_C$s than those of the primary OA factors (Table 3). Flaming-BBOA
comprised 76 % and 55 % of the unaged OA from dry and moist spruce combustion in the masonry heater, respectively, but
only 27 % and 23 % in the beech and moist spruce combustion OA from the chimney stove, respectively (Fig. 10), indicating
a less-oxidising higher-temperature flaming phase in the modern masonry heater. Flaming-BBOA is strongly related to the
$C_4H_9^+$ -ion (main peak of m/z 57, $R^2 = 0.90$, Fig. S18) which is used as a tracer for hydrocarbon-like compounds (Aiken et al.,
2009). However, in contrast to the typical hydrocarbon-like OA factor characterised by a relatively low $OS_C$ (-1.7 to -1.6, Kroll
et al., 2011), the flaming-BBOA contains more oxygen-containing functional groups, and is similar to the primary biomass
burning OA factor measured in the RWC exhaust in a chamber (Tiitta et al., 2016; θ = 13.6º, $R^2 = 0.95$). Pyrolysis-BBOA, on
the other hand, consisted of ions typical to the low-temperature pyrolysis products of wood combustion and correlated well
with the PACs ($R^2 = 0.86$, Fig. S18).

Of the more-oxygenated factors, the LV-OOA was dominated by the $CO^+$ and $CO_2^+$ ions and thus represented highly
oxidised OA. The LV-OOA spectra corresponded well with the OH-induced SOA factor identified from the RWC exhaust
aged in a smog chamber (Tiitta et al., 2016., θ = 8.4º, $R^2 = 0.98$), and was comparable with that of ambient LV-OOA (Aiken et
al., 2009) and the LV-OOA spectra of unaged wood combustion exhaust (Kortelainen et al., 2018). LV-OOA was also present
in the unaged exhaust of this study, excluding the dry spruce combustion in the masonry heater, which produced the least-
oxidised primary exhaust. The SV-OOA, on the other hand, was related to the $C_2H_3O^+$-ion ($R^2 = 0.78$), which is indicative of
carbonyl formation in the particulate OA (Ng et al., 2010). The SV-OOA was also comparable (θ = 11.5 º, $R^2 = 0.96$) to a
factor of SOA generated in a chamber in the previous work (Tiitta et al., 2016) where this factor was  interpreted as ozonolysis-
generated organic OA. Interestingly, SV-OOA was formed only during photochemical aging and increased in line with higher
exposures despite semi-volatile compounds being products of the initial stages of OA oxidation. This further demonstrates the
long-continuing functionalisation of OA during aging alongside with the fragmentation processes.

### 3.5.3 Polycyclic aromatic compounds

PACs were overall more prominent in the primary exhaust of dry spruce combustion (3.2 % of total OA, Fig. 11) than in moist
spruce (1.7 % or 2.0 %) or beech (1.7 %). As expected, the total PAC concentrations decreased because of aging and
contributed less than 0.5 % to OA for all cases after 3 eqv.d of aging. Furthermore, aging transformed the composition of the





PACs assessed with the AMS HR-PAH analysis (P-MPIP analysis, Chapter S5.2; Herring et al., 2015). While UnSubPAHs formed the most prominent PAC group of all the combustion experiments, aging decreased their share from 60 % in unaged exhaust to 40–50 % after 3 eqv.d of aging. Similarly to the PACs measured with AMS, aging decreased the UnSubPAHs

analysed with IDTD-GC-ToFMS by 83–85 % in the dry spruce combustion in the masonry heater, and by 90–98 % in the other situations. Of the most prominent UnSubPAHs, anthracene and fluoranthene degraded within the first eqv.d (Fig. 12). Also benzo[a]pyrene, which is a used as the marker for total ambient PAHs (EC 2004), degraded by a factor of 5 because of aging (Table S12).

The decrease in total PAC concentration likely diminished the adverse health effects of the exhaust; however, there

was a simultaneous formation of oxy- and nitro-PAC derivatives known to be detrimental to health. These substituted PACs have lower vapour pressures compared to those of parent PAHs and thus are more likely to condense on the particles (Shen et al., 2012). The share of oxygenated PACs (OPAHs) to the total HR-PAH concentration measured by AMS increased from 15–19 % in unaged exhaust to 25–38 % in aged exhaust. The concentrations of both the AMS HR-OPAH and IDTD-GC-ToFMS derived OH-PAH also correlated with the SV-OOA PMF factor (Pearson r = 0.70 for OPAH and r = 0.88 for OH-PAH; see

Table S11) pointing towards their continuous formation during aging. Of the compounds measured by IDTD-GC-ToFMS, the most notable increase was observed for naphthaldehydic acid, with high concentrations (up to 100 µg m⁻³) in aged aerosol. Interestingly, its concentration was highest at approximately 1 eqv.d, after which it decreased. Naphthaldehydic acid and other oxygenated PAHs have also been previously found to form during photochemical aging of RWC exhaust (Bruns et al., 2015; Miersch et al., 2019); however, we found that the photochemically enhanced naphthaldehydic acid concentration degraded

after continuous aging, although remaining considerably higher than those in unaged emissions.

In addition, the share of nitrogen-substituted PAHs, including both NPAH and APAH, increased from a combined share of 5 % of HR-PAH$_{tot}$ in fresh exhaust to a maximum of 9 % in the aged exhaust. In general, particulate nitrogen-substituted PAHs are formed in the atmosphere through the oxidation of gaseous PACs or via heterogenous reactions from UnSubPAHs and are also simultaneously degraded by photochemical reactions (Keyte et al., 2013). As UnSubPAHs in the

present study were largely consumed after 3 eqv.d, the higher photochemical exposure times consequently led to a decrease in nitrogen-substituted PAHs.

### 3.5.4 Organic acids

Photochemical aging led to a considerable increase in particulate organic acids in exhaust aerosol. The amount of small multifunctional acids such as malic acid ($C_4H_4O_4$) and tartaric acid ($C_4H_6O_6$) increased by factors up to greater than 200 (Fig.

12). Increases in the amounts of also dicarboxylic acids such as succinic ($C_4H_6O_4$) and glutaric acid ($C_5H_8O_4$) were evident, although to a lesser factor. Compared to the AMS measurements, the concentrations of organic acids were connected to both LV- and SV-OOA factors (r > 0.7, Table S11), which also strongly increased during the aging process. This was expected based on the association of SV-OOA with the $C_2H_3O^+$-ion indicating carbonyl formation. In the $n_C$-OS$_C$ space (Fig. 9), these compounds dominated the upper-right corner with the highest oxidation states and largest enhancement ratios and are close to





the LV-OOA region specified for atmospheric OA by Kroll et al. (2011). These organic acids were intermediately volatile products of the continuous fragmentation process and partitioned also to the gaseous phase where they in turn participated in secondary gas-phase OH reactions. Their concentrations were highest at approximately 3 eqv.d, after which they were degraded by further oxidation reactions.

### 3.5.5 Nitrophenols

An increase in nitrophenols in the particulate phase was evident during photochemical aging, which increased the concentrations of 4-nitrophenol (4-NP) and 4-nitrocatechol (4-NC) by respective factors of 2–30 and 30–3000 compared to non-aged exhaust (Fig. 14). The highest attained 4-NC concentration corresponded to 2 % of total AMS OA concentration, which is a notable fraction for a single compound in an aging aerosol. These secondary nitrophenols were products of OH + phenolic-compound reactions and may have originated from both the gas-phase and heterogenous reactions (Harrison et al.,

2005). However, ozonolysis in the presence of $NO_2$ led to an extensive formation of 4-NP, whereas the amount of 4-NC did not increase. This discrepancy might be the result of absence of the photochemical production of catechol, which is the precursor for 4-NC formation (Finewax et al., 2018).

       Nitrophenol concentrations were highest at relatively low OH exposures (1–2 eqv.d) and decreased with increased aging. Similar trends with OH exposure were seen in the NPAH and APAH concentrations measured with AMS. Nitrophenols

are reactive towards OH and photolysis in both gaseous and particulate phases, and as their amount decreased from both phases after the first equivalent days of aging, it can be concluded that they overall diminished from the exhaust with long-term aging. However, their concentrations did remain higher in the highly aged exhaust than in the fresh exhaust. Furthermore, in the particulate phase, they likely contributed to the formation of organic acids which are formed during the continuous photo-oxidation of the nitrophenols (Hems and Abbatt, 2018).

### 4 Conclusions


The photochemical aging of dynamically changing OA emitted from RWC was evaluated using the PEAR oxidation flow reactor to expose the exhaust to varying photochemical conditions for up to an equivalent of one week in the atmosphere. To evaluate typical Northern and Central European combustion emissions, two different appliances were used with regionally typical logwood fuels. While the primary concentrations of particulate OA were relatively similar for all of the assessed

sources, the enhancement of the organic particulate carbon during aging depended on the type of the fuel. In particular, the moisture content affected the SOA production with dry fuel producing a smaller organic mass during aging because the OA enhancement was strongly influenced by the emission rates of the organic gases, which were significantly lower for dry spruce. However, a very low (5 %) moisture content also considerably increased the primary $PM_1$ emission because of the extensive soot formation. With current logwood combustion appliances, this presents a conflict in emission reduction intended to



decrease the different constituents in RWC exhaust with overly dry logwood producing large black carbon emissions and moist fuel increasing oxygenated emissions.

The particulate organic carbon mass in the RWC exhaust increased by a factor of 1.3–3.9 during photochemical aging. Furthermore, photochemical aging transformed the overall composition of the OAs. This was observed as a linear increase in the average carbon oxidation state of particulate OA throughout the investigated photochemical exposure range, while the ratio

of organic carbon to total organic mass decreased. Photochemical aging caused multiform changes in the OA also at the molecular level. Notably, small, acidic reaction and fragmentation products became increasingly dominant in both particulate and gaseous phases with higher aging. The concentrations of particulate nitrophenols were at their highest level after 1 eqv.d, after which they began to decaying but remained higher than that in the primary exhaust. Similarly, nitrogen-containing aromatics were unobservable in the gaseous phase at longer exposures, although they have been observed to increase during

exposures less than 1 eqv.d (Hartikainen et al., 2018). Aging also enhanced the share of oxygen- and nitrogen-substituted polycyclic aromatic compounds in the PAC emissions. Of the oxygenated PACs, naphthaldehydic acid in particular increased considerably with concentrations peaking at approximately 1 eqv.d. However, PACs in total degraded almost completely after 3 eqv.d of aging.

Based on this work, different transformation pathways for RWC exhaust under photochemical conditions can be

roughly outlined: the initial pathways consisting of functionalisation and condensation from gaseous precursors are followed by more particulate-phase-driven chemistry consisting of heterogeneous oxidation and fragmentation. While several recent studies (Bertrand et al., 2017; Bruns et al., 2015; Grieshop et al., 2009; Heringa et al., 2011; Tiitta et al., 2016) assessed the behaviour of RWC emissions in a timescale of less than 1.5 eqv.d, the results of this study emphasise the importance of investigating longer photochemical exposures. This is particularly relevant considering the potentially long atmospheric

lifetimes of OA and its importance in large-scale atmospheric models which typically estimate SOA formation and characteristics based on short-term chamber experiments. Atmospheric OA is a mixture of emissions from various sources having diverse exposure levels from fresh emissions to long-transported highly oxidised OAs. The potential health and climate effects of aerosols are to a large extent determined by their composition, which depends on their sources and the levels of atmospheric aging. Thus, the characterisation of aerosol emissions from different sources and their atmospheric transformation

at different exposure levels is crucial when assessing the overall environmental effects of ambient air pollution.

**Data availability**

The data is available on request from the corresponding author.



## Author contributions

AH, OS, PYP, MI, and HL designed the study. Measurements were performed by AH, PT, MI, PYP, MK, HL, HS, JT, and
OS. AH, HZ, MI, and OS made the assessments of the PEAR conditions. AH performed analyses of FTIR, SMPS, and PTR-
ToF-MS data, PT and LH performed the AMS data analyses, JO performed IDTD-GC-ToFMS analyses, and HK performed
the thermo-optical analyses. OS, JT, RZ, and JJ supervised and acquired funding the study. Paper was written by AH with
contribution from all the co-authors.

## Competing interests

The authors declare that they have no conflict of interest.

## Acknowledgements

We thank Donna Sueper from Aerodyne Research, Inc, for her work on the AMS HR-PAH analysis. Financial support by the
Academy of Finland (ASTRO-project (Grant 304459) and NABCEA-project (Grant 296645)), Doctoral School of University
of Eastern Finland, German Research Foundation grant ZI 764/14-1, and the Helmholtz virtual Institute HICE (www.hice-
vi.eu; InhaleHICE) and the aeroHEALTH Helmholtz International Lab. (www.aerohealth.eu) is gratefully acknowledged.

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



**Table 1: Combustion conditions and experimental conditions of each experiment.**

|  | Exp. | $h\nu$ flux [photons cm$^{-2}$ s$^{-1}$] | O$_3$ input [ppm] | Initial age* [eqv. d] | Fuel | OH exp. [# s cm$^{-3}$] | Age [eqv.d] | MCE | VOC/NO$_x$ | DR |
|---|---|---|---|---|---|---|---|---|---|---|
| Masonry heater | 1 | lamps off | 0 | no-OH | S-5% |  |  | 0.979 | 1.5 | 64 |
|  |  |  |  |  | S-22% |  |  | 0.974 | 4.9 | 83 |
|  | 2 | 8.3E+15 | 2.3 | 14.07 | S-5% | 5.8E+11 | 6.7 | 0.975 | 1.2 | 137 |
|  |  |  |  |  | S-22% | 3.5E+11 | 4.0 | 0.978 | 3.1 | 147 |
|  | 3 | lamps off | 2.5 | no-OH | S-5% |  |  | 0.974 | 1.3 | 135 |
|  |  |  |  |  | S-22% |  |  | 0.973 | 4.0 | 144 |
|  | 4 | 1.9E+15 | 2.2 | 5.84 | S-5% | 1.5E+11 | 1.7 | 0.974 | 1.4 | 130 |
|  |  |  |  |  | S-22% | 1.3E+11 | 1.5 | 0.966 | 5.5 | 152 |
|  | 5 | 1.4E+15 | 1.8 | 2.35 | S-5% | 8.1E+10 | 0.9 | 0.971 | 1.3 | 160 |
|  |  |  |  |  | S-22% | 6.3E+10 | 0.7 | 0.976 | 3.7 | 196 |
| Chimney stove | 1 | lamps off | 0 | no-OH | B-17% |  |  | 0.968 | 1.6 | 72 |
|  |  |  |  |  | S-22% |  |  | 0.966 | 5.4 | 80 |
|  | 2 | 5.4E+15 | 4.3 | 12.90 | B-17% | 3.1E+11 | 3.6 | 0.963 | 1.8 | 122 |
|  |  |  |  |  | S-22% | not meas. | ~4** | 0.965 | 3.6 | 148 |
|  | 3 | lamps off | 3.1 | no-OH | B-17% |  |  | 0.953 | 1.9 | 124 |
|  |  |  |  |  | S-22% |  |  | 0.961 | 4.9 | 143 |
|  | 4 | 1.1E+15 | 3.6 | 5.10 | B-17% | 5.4E+10 | 0.6 | 0.951 | 1.7 | 119 |
|  |  |  |  |  | S-22% | 6.1E+10 | 0.7 | 0.972 | 4.2 | 135 |
|  | 5 | 1.1E+16 | 11 | 18.76 | B-17% | 5.0E+11 | 5.8 | 0.962 | 1.5 | 36 |
|  |  |  |  |  | S-22% | 5.0E+11 | 5.8 | 0.957 | 5.6 | 40 |

*Age based on the OH-exposure of clean air, prior to sample input.
** Direct OH exposure measurements not available for Chimney stove Exp. 2 S-22%; approximately similar as in Exp. 2 B-17%.

**Table 2: Primary PM$_1$ emissions in dry 13% O$_2$ exhaust. The number concentration and geometric mean mobility diameter (GMD) were derived from a scanning mobility particle sizer and PM1, organic carbon, and elemental carbon from filter samples.**

|  | Number (10$^7$ # cm$^{-3}$) | GMD (nm) | PM$_1$ (mg m$^{-3}$) | OC (mg m$^{-3}$) | EC (mg m$^{-3}$) | OC:EC |
|---|---|---|---|---|---|---|
| Masonry heater, spruce 5 % H$_2$O (S-5%) | 3.2 ± 0.6 | 95.5 ± 24.4 | 67 ± 16 | 4.3 ± 1.9 | 57.1 ± 15.8 | 0.07 ± 0.02 |
| Masonry heater, spruce 22 % H$_2$O (S-22%) | 4.5 ± 1.6 | 68.4 ± 22.2 | 33 ± 16 | 4.4 ± 4.6 | 19.1 ± 12 | 0.31 ± 0.45 |
| Chimney stove, beech 17 % H$_2$O (B-17%) | 4.4 ± 0.4 | 61.1 ± 7.6 | 43 ± 11 | 3 ± 1.4 | 21.9 ± 11.1 | 0.15 ± 0.04 |
| B-17% : 1st to 2nd batch | 4.3 ± 1.9 | 71.6 ± 5.7 | 48 ± 4 | 4.2 ± 0.9 | 32.0 ± 5.3 | 0.12 ± 0.01 |
| B-17% : 3rd to 4th batch | 5.4 ± 0.9 | 57.2 ± 5 | 38 ± 13 | 1.9 ± 0.6 | 11.8 ± 3.3 | 0.18 ± 0.03 |
| Chimney stove, spruce 22 % H$_2$O (S-22%) | 4.5 ± 0.3 | 52.8 ± 13.3 | 37 ± 8 | 4.3 ± 0.7 | 17.0 ± 2.0 | 0.25 ± 0.04 |






**Table 3: Properties of the PMF factors.**

| Factor | O:C | H:C | N:C | OS$_C$ | OM:OC |
|---|---|---|---|---|---|
| LV-OOA | 1.57 | 0.97 | 5.97E-03 | 2.18 | 3.19 |
| SV-OOA | 0.78 | 1.41 | 1.78E-03 | 0.16 | 2.17 |
| Flaming-BBOA | 0.55 | 1.52 | 4.33E-03 | -0.42 | 1.86 |
| Pyrolysis-BBOA | 0.49 | 1.41 | 2.13E-03 | -0.44 | 1.77 |

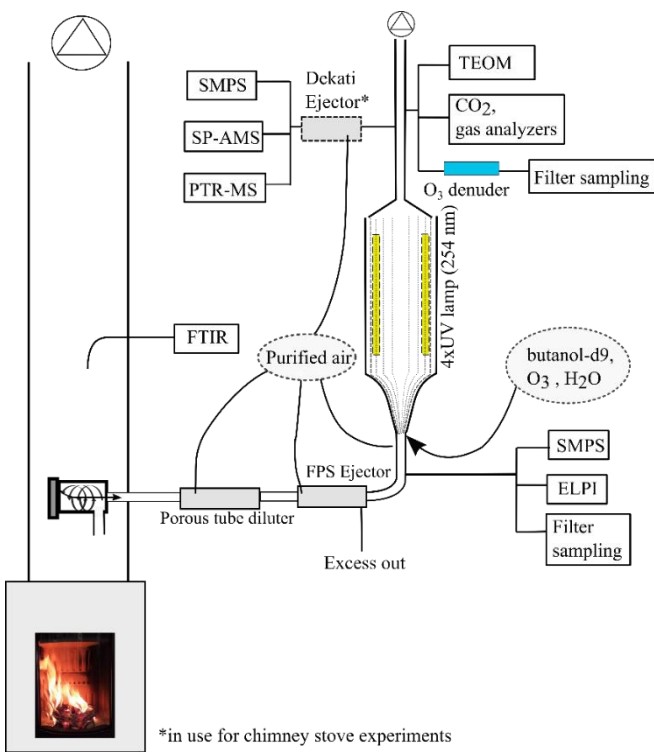

**Figure 1: Experimental setup.**





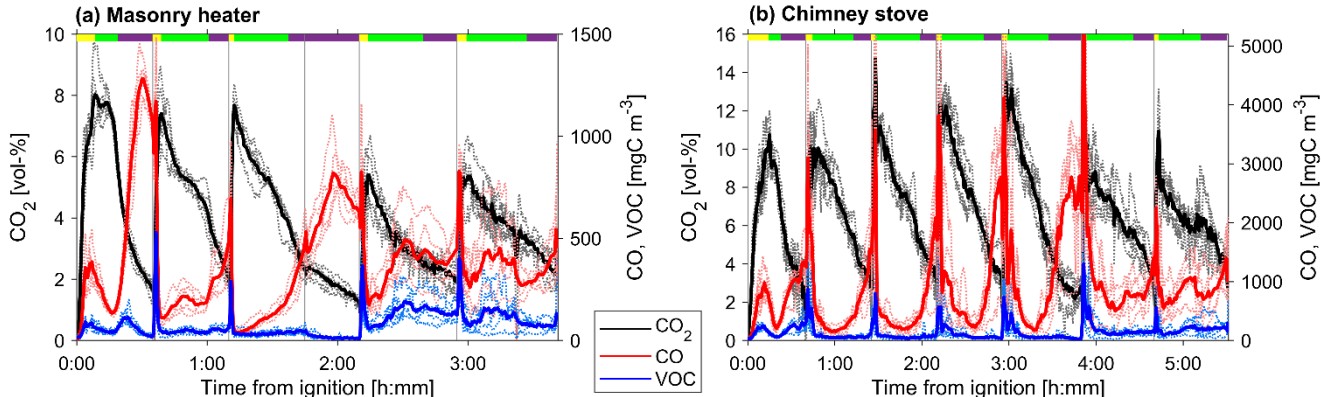

**Figure 2: CO₂, and CO, and the VOCs in the exhaust gas from (a) masonry heater, and (b) chimney stove, measured by FTIR. Averages over all experiments are shown with solid lines, whereas different experiments are shown with dotted lines. Average phase lengths are marked on the top panel with the yellow (ignition), green (flaming), and purple (burnout).**

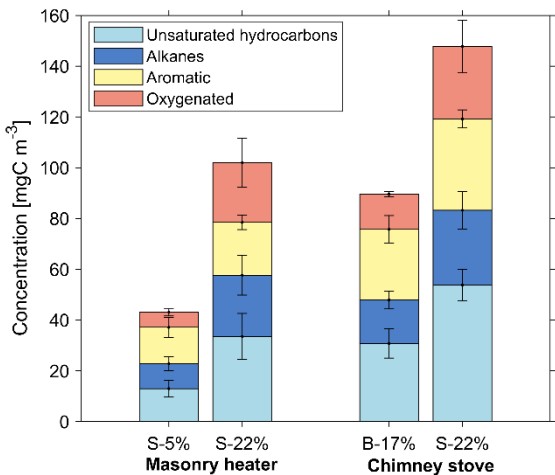


**Figure 3: Average concentrations of organic gaseous compounds in the primary exhaust measured by FTIR from the stack. Error bars denote standard deviations between experiments.**





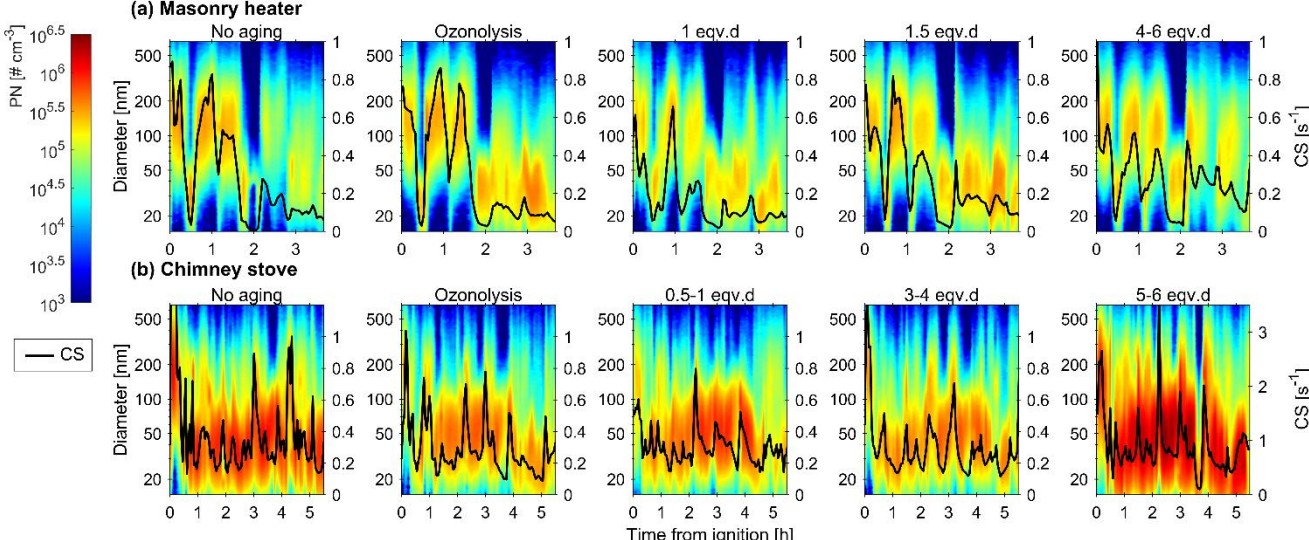

Figure 4: Size distributions of diluted primary aerosol entering the oxidation flow reactor as a function of time, and condensation sinks (CS) in the PEAR based on average of size distributions before and after PEAR.

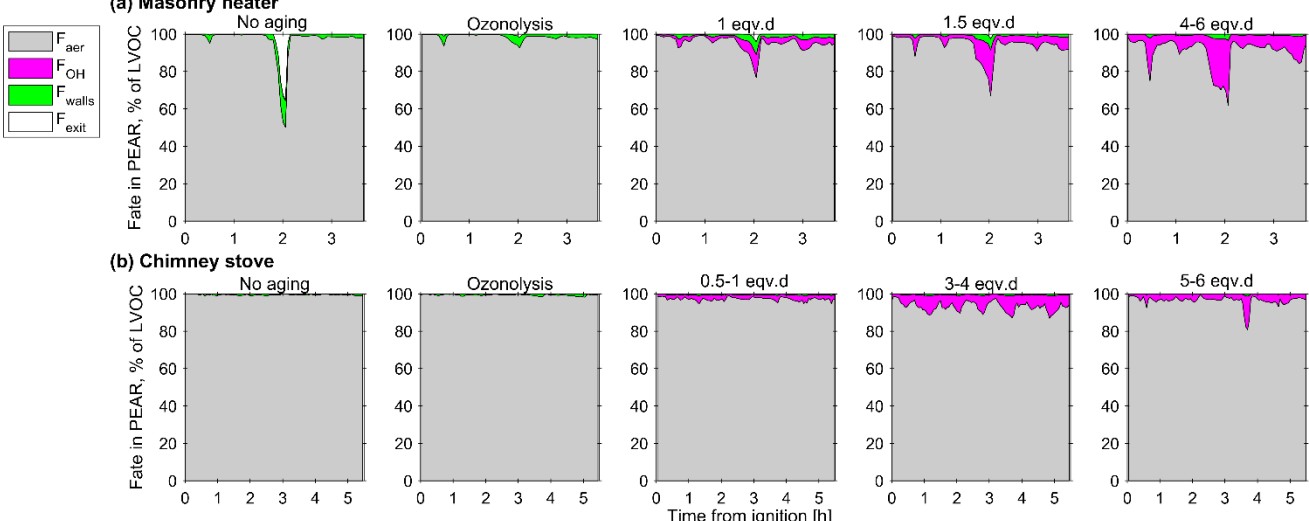

**Figure 5: Estimated fates of the low-volatility organic compounds in the PEAR during the experiments with portions of the LVOCs condensing onto particles (F$_{aer}$) and walls (F$_{walls}$) and lost in reactions with OH (F$_{OH}$). The remainder of the LVOCs (F$_{exit}$) exit the PEAR as LVOCs.**





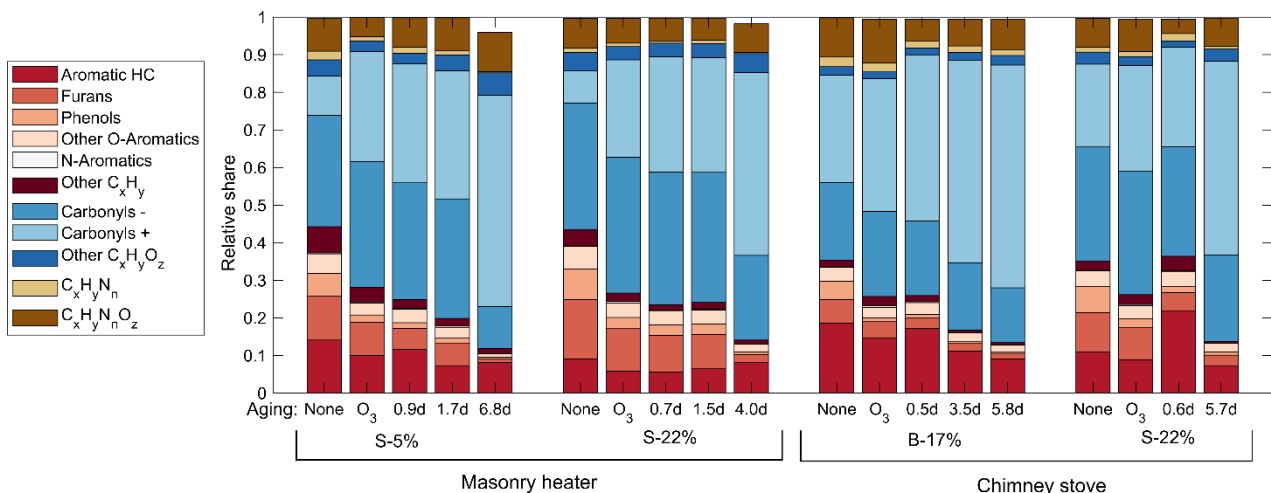

**Figure 6: Relative shares of the identified VOC groups in the exhaust after PEAR as measured by PTR-ToF-MS.**

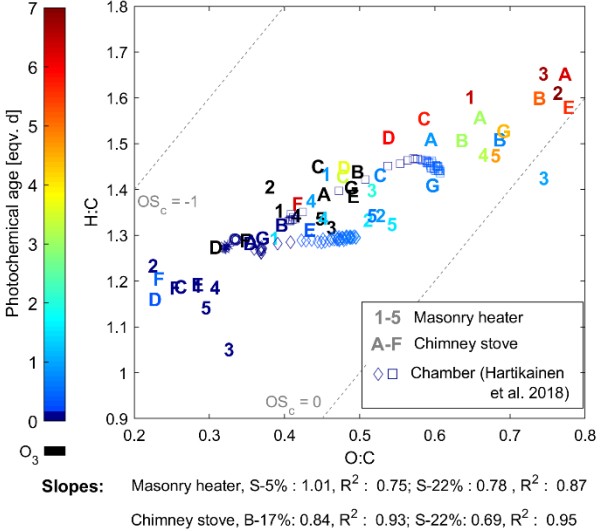

**Figure 7: Van Krevelen diagram of the VOCs measured with PTR-ToF-MS. Numbers indicate batches combusted in masonry heater (1–3 dry spruce, 4–5 moist spruce); letters refer to chimney stove batches (A–E beech, F–G moist spruce). Ozonolysis experiments (in black) were not considered in the slope calculations. Behaviour of RWC emission aged in a chamber (Hartikainen et al., 2018) is marked with scatter.**





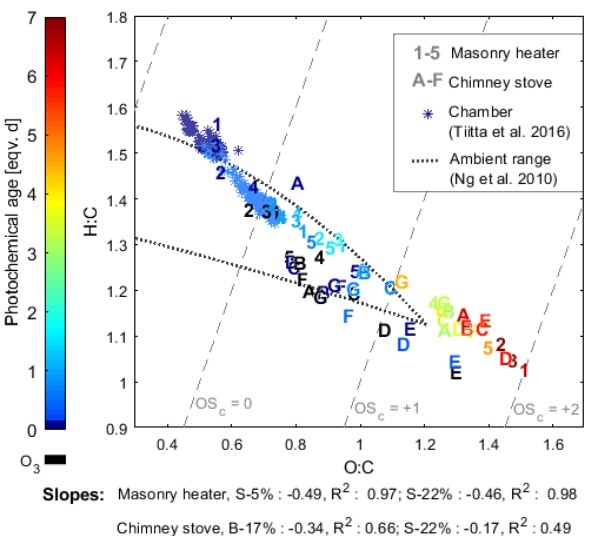

**Figure 8: Van Krevelen diagram of the particulate aerosol measured by AMS. Numbers indicate batches combusted in masonry heater (1–3 dry spruce, 4–5 moist spruce), whereas letters refer to chimney stove batches (A–E beech, F–G moist spruce). Ozonolysis experiments are marked with black and were not considered in slope calculations. The behaviour of RWC OA aged in a chamber (Tiitta et al., 2016) with slopes of -0.64 to -0.67 is marked with scatter.**

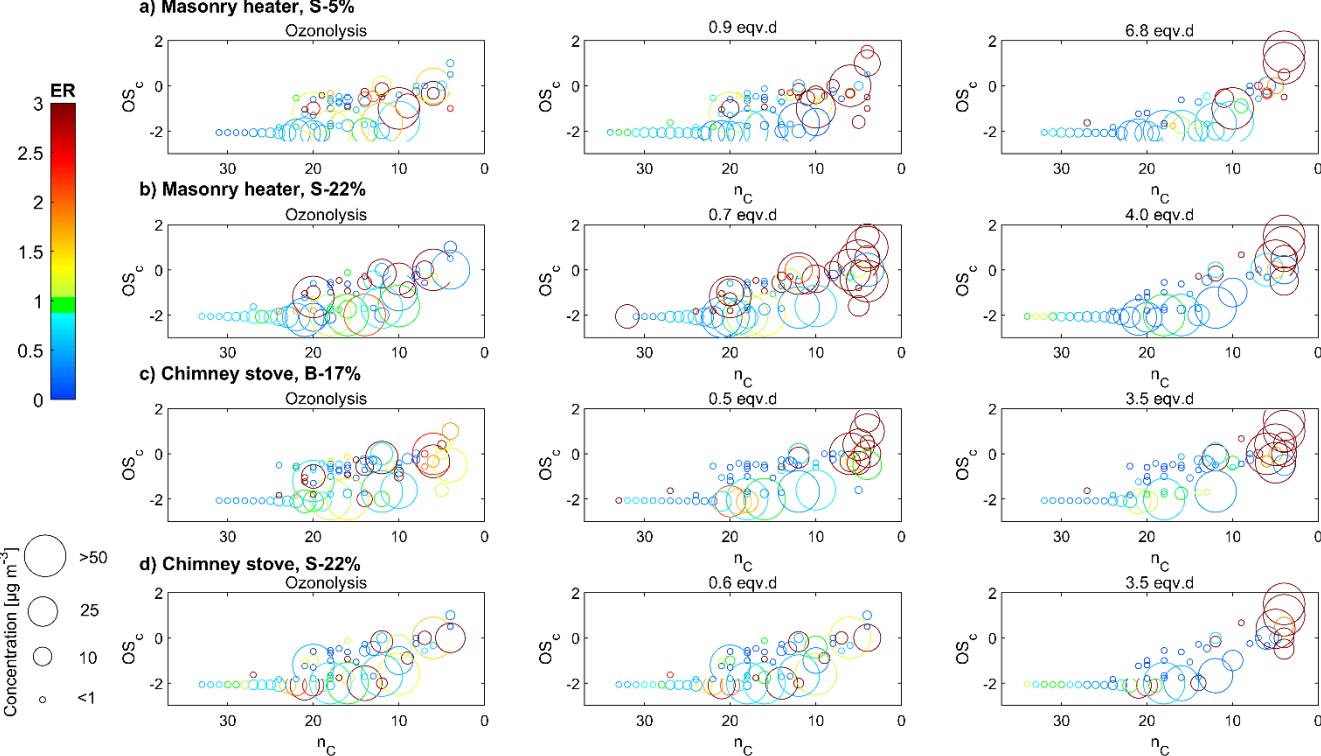



**Figure 9: Compounds measured by IDTD-GC-ToFMS in different experiments (ozonolysis, low, and high photochemical exposure) with respect to their carbon number ($n_C$) and oxidation state ($OS_C$). Enhancement ratios (ER) compared to the experiments without oxidative aging are shown in colour, and the size indicates the dilution-corrected concentrations in the secondary exhaust.**

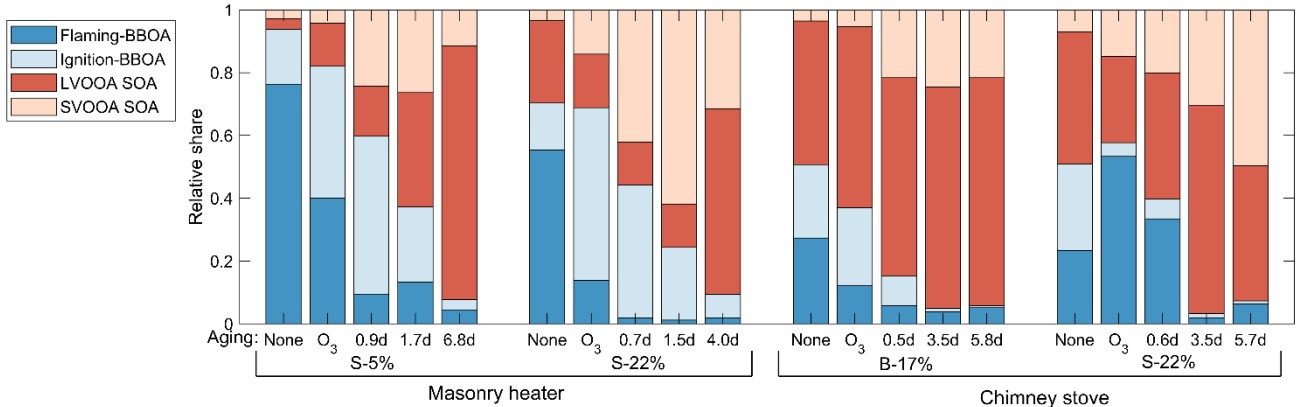

**Figure 10: Average shares of the four PMF factors in the exhaust after the PEAR at the different exposure levels.**

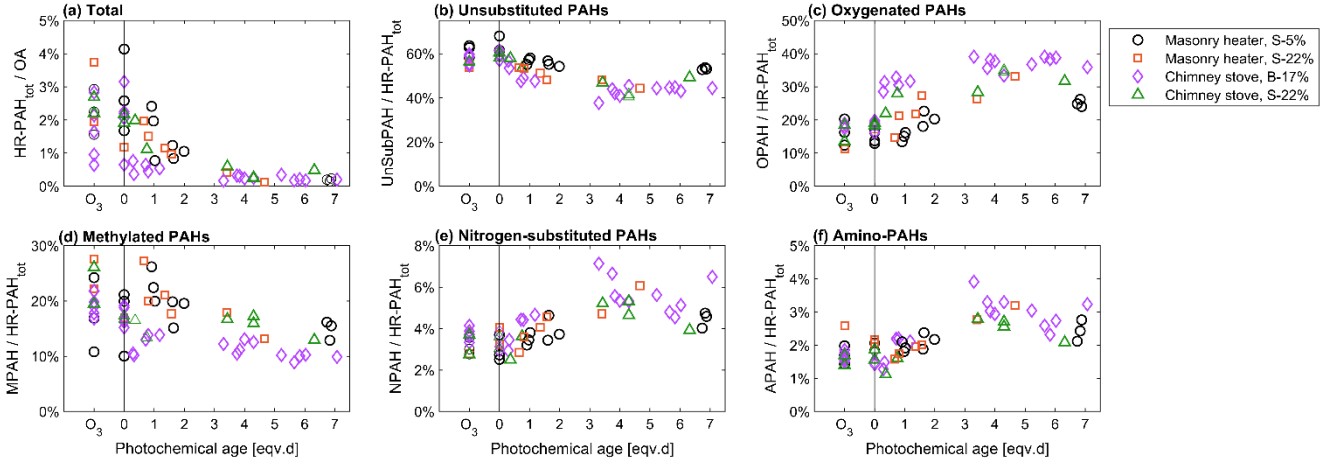

**Figure 11: Relationship of photochemical aging to the batchwise average ratios of: (a) total PAH concentration to OA concentration measured by AMS, and (b–f) of HR-PAH subgroups to the total PAC concentration.**





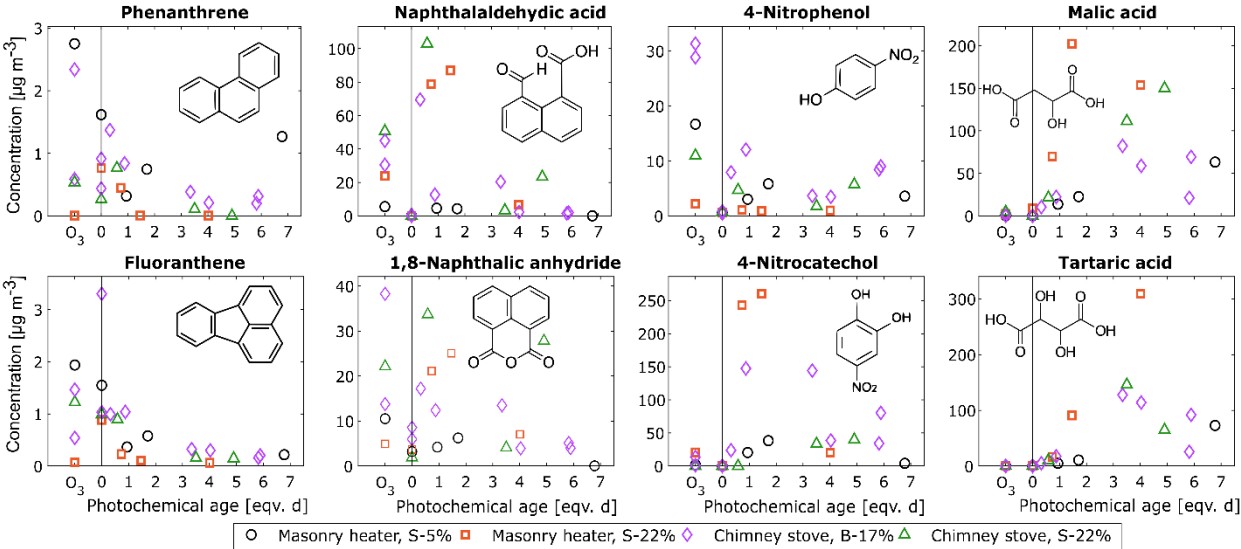

**Figure 12: Concentrations of selected compounds measured by IDTD-GC-ToFMS at different exposure levels.**