# Peer review of "Photochemical transformation of residential wood combustion emissions: dependence of organic aerosol composition on OH exposure"

_Atmospheric Chemistry and Physics, 2019_

## Referee Comment (RC1) · Anonymous Referee #1 · 3 Jan 2020

GENERAL COMMENTS:

This article adresses photochemical aging of residential wood combustion (RWC) emission from two stoves at the home-built PEAR oxidatation flow reactor, reaching OH exposures representative of aging of up to one week in the atmosphere. This is one of few studies adressing the topic of RWC aging in OFRs with a suite of state-of-the-art mass spectrometric online and offline techniques and adds value for the scientific community.

[Figure]

In my view, the manuscript is of high scientific quality and presents relevant results from a comprehensive study. The language and structure, however, lack clarity and some information is re-dundant which makes the manuscript unnecessarily long and the core findings quite dilute. I suggest that this be improved to make the article more concise and easier to follow and understand for the readers. I believe this is feasible with major revisions oft text and paragraph structure in order to allow more clarity on the main findings, conclusions and limitations of the study. Aside of specific com-ments below, re-formatting of units, table layouts, etc., into one consistent format and re-structuring paragraphs may help to achieve this. As a general comment, the results section would benefit if the authors would focus on summarizing their main findings first, and dis-cuss the limitations of the study in a second stage with a focus on their implications of the obtained results, rather than first discussing experimental conditions/limitations as results followed by reporting the actual findings.

In summary, I have enjoyed seeing these results and am looking forward to see the revised manuscript published.

SPECIFIC COMMENTS:

Overall:

o The main findings are dilute throughout the manuscript and difficult to grasp. Parts in the conclu-sion section remain vague too. I suggest to summarize the main goals and questions of the stu-dy as specific as possible at the end of the introduction and to state the core findings as specific as possible in the conclusions section.

o While the manuscript title is focused primarily on the photochemical aging of the emissions, a substantial part of the manuscript discusses primary emissions (Fig 2+3), operating/exerpimental conditions (Fig 4+5) and only the second part of figures (Fig 6-11) and results actually adresses the aging and chemistry question. In my view, some of the initial figures (e.g. 4+5) could be mo-ved to the SI along with their text in favor of clarity of the manuscript; the implications of the de-termined losses in Fig 4+5 should

then be referred to discussing the actual results, i.e. for examp-le, the authors might adress how do particle/vapor losses influence the observed decay of com-pounds as a function of OH exposure in their experiments. Alternatively, Figure S1, S3 and S4 could fit well with the main text, as they summarize the primary emissions composition, which is relevant for any following discussion of emissinos aging. Further remarks to help clarity and focus are listed later on.

Abstract, Introduction & Methods:

o L20/ Abstract: suggest to specify here which oxidation flow reactor is used in this study; suggest to use "PEAR OFR" throughout

o L21 vs L 28 and other occasions/ Abstract: The authors mix between "gaseous organic com-pounds" and "volatile organic compounds"; suggest they try to harmonize the language

o L27/ Abstract: specify based on which analysis the acetic fragmentation is investi-gated

o L30 / Abstract and other occasions: the authors use the more generic term "polycyclic aromatic compounds (PACs)" rather than polycyclic aromatic hydrocarbons (PAHs) which has been used in previous related work and is also mentioned in e.g., L75. I suggest to specify and/or define for non-specialists, why the authors refer to PACs in some cases but PAHs in others. This should also take into account the limitations of their instrumentation to effectively distinguish between PACs and PAHs. In my un-derstanding, frequently only fragments of PACs can be detected with PTR-ToF-MS or AMS, however, they are assumed to be and referred to as PAHs in the manuscript.

o L36 / Abstract: suggest to specify which scale is "fresh", "shortly aged", and "long-term aged" in the context of this study

o L54: Is there a difference between "fresh" and "unaged"? Otherwise, I suggest the authors rephrase to read "fresh, i.e. not atmospherically aged"

o L60/61: The authors may include in this section of SOA-precursor discussion that removing aro-matic hydrocarbons from wood burning emissions by use of e.g. catalytic converters can drasti-cally reduce SOA formation, as recently presented by Pieber et al., 2018; also changing applicance operation and modifying combustion phases to conditions that emit less aromatic hydrocarbons might have this effect; Figure S3+S4 are valuable indicators in this context.

o L87: suggest to provide some references for previous RWC OFR studies in addition to the men-tioned smog chamber experiments; e.g. Bruns et al. 2015, Czeck et al. 2017, Pieber et al. 2018, etc.

o L132: "In addition"... "additional"; language is redundant. Suggest to remove one of the two.

o L133-134: Ozone and butanol-d9 metrics in volumetric flows does not provide any information on concentration levels; could the authors in addition or instead specify the mixing ratios?

o L134: "formed" should probably read "forms"

o L135: "depended" should probably read "depends"

o L135: "H2O" should probably read "H2O vapor"

o L143: "OH concentration" should read "24-hour average global OH concentration"

o L146: was butanol-d9 mixing ratio included in this equation an what is ist relative contribution to the total OHR external?

o L148: suggest the authors present the main results of OHR external analysis in one brief sentence in this paragraph, and mention its implications for the results and conclusions later on.

o L150ff: suggest the author present the main results of particle and LVOC loss esti-mates in 1-2 brief sentences in the main text here and move the remaining text on this

topic from the results section to the SI, discussing it in the main text only as a limitation oft e study including its impli-cations for conclusions (as reasoned above). I suggest also to define what LVOCs refers to in the context of this article.

o Line 160/161: I suggest to keep all information in one paragraph, rather than starting a new para-graph here.

o L167: suggest to define semi-VOCs in this context.

o L168: "isotope labelled" should probably read "isotopically labelled"

o L173: suggest to replace "after the PEAR" with "after the PEAR, i.e. at its outlet"

o L187: the author refer to this instrument as "AMS" throughout the manuscript; I suggest to chan-ge this to SP-HR-ToF-AMS on all occasions, starting from the abstract, as is done for all other in-strumentation (e.g. PTR-ToF-MS).

o L189: Why did the authors choose to use the "Improved-Ambient method" for this laboratory da-ta set?

o L200: Suggest to compare also with Bruns et al., 2015 who have used this method beforehand for an intercomparison of spectra from OFR and smog chambers including RWC.

o L205: Suggest to replace "residential wood combustion" with RWC

o L209 (from L186 onwards): as noted in the next comments; please add important information on AMS data anylsis in the main text.

o Some text is repetitive or split into different locations (partially found in main text, partially in duplicate SI), while other text is grouped together into subchapters which are not entirely logical. I suggest to make the overall language more concise and shorten the text, while keeping all rele-vant information in the main text. E.g. some suggestions:

• L170 (main text): I suggest to relabel this as "Online aerosol particle and gasphase measu-rements" to avoid ambiguity." It may be beneficial to split the subsection 2.4 into three sections for each instrument, and then make further separat paragraphs for any AMS in-formation aggregated as a) experimental/calibration/raw data correction, b) PMF analysis, c) HR fitting or PAH fitting.

• L186 onwards (main text): relevant AMS operating conditions (e.g. size-cut off) and data corrections (RIE=1.4 for OA, CO2+-interference correction, CO2 gasphase correction) should be mentioned in the main text; I suggest to move information from SI (Section S5) entirely to the main text, but shorten the paragraph by avoiding redun-dencies (e.g. "The AMS data was analysed using the standard analysis tools SQUIR-REL v1.62A and PIKA v1.22D adapted in Igor Pro 8 (Wavemetrics)." is currently stated double. With regards to the RIE=1.4, please mention that this is valid for OA. With regards to the CO2+-interference correction, please mention the magnitude of inter-ference and potential im-pacts on the determined O/C ratios. What was the level of inorganic nitrate to OA in the primary and secondary emissions? With regards to CO2 gasphase-correction, please men-tion whether this was done by standard measure-ments of particle-filtered air during the experiment or external calibration with gaseous CO2.

• L190 onwards (main text): I suggest to make a new paragraph with all information rele-vant to PMF, see comments above.

Results:

o L210: Suggest to relabel as "results and discussion", given the separate section entitled "conclusi-ons" in L540

o Section 3.1.: Is there any information on the temperatures during these different combustion pha-ses available?

o L228: "constantly" should probably read "continuously"; "from the diluted flue gas"

should pro-bably read "in the diluted flue gas".

o L232: "primary aerosol" is certainly technically correct if referring to aerosol as particles plus their surrounding gas, however, it may quickly become confusing as often "aerosol" is used when refer-ring to the particle phase only. I suggest to replace "primary aerosol" with "primary emissions" here.

o L233: suggest to specify the value of protonation efficiency (proton affinity) here

o L230: suggest to state the limitations of both, FTIR and PTR-ToF-MS, here briefly, e.g. for PTR-ToF-MS, rather than point out alkanes specifically, it should be noted that any molecules with proton affinity lower than that of water (in H3O+ mode) can not be protonated and hence detec-ted; further, I suggest to mention based on which criteria 127 molecular formulas were identified and how many ions remained unidentified. For the FTIR, it would be informative to give a brief reasoning why only 27 compounds can be detected and quantified, rather than only refer to the table in SI.

o L233: fragmentation does not necessarily limit the quantification but rather the iden-tification; suggest to rephrase

o L236: suggest to introduce the later used abbreviation "ArHC" here at its first occa-sion

o L243: suggest to replace "significant" by "statistically significant" here

o L259: ArHC are also discussed earlier, but the abbreviation is only introduced in L259; suggest to revise and use and define ArHC earlier on

o L265: Suggest to add that previous studies (e.g. Pieber et al. 2018), demonstrated that removing ArHC from the emissions mix substantially reduces their SOA formation.

o Figure 7 and 8: where do O/C and H/C ratios of other wood burning OFR studies fall in compari-son to the obtained results (e.g. add data from Bruns et al., 2015, Czech et al. 2017, Pieber et al., 2018, and similar data from other research groups as available).

o L371: Replace "consequent" with "subsequent"

Conclusions

o L556: "Notably, small, acidic" should probably read "Notably, small acidic" (i.e. without comma)

o L565: given that the presented manuscript discusses OFR-experiments, I suggest to cite and dis-cuss also other OFR studies with RWC rather than only smog chamber experiments; e.g. Czech et al. 2017, Bruns et al. 2015, Pieber et al. 2018: how do those PAM OFR studies compare in OH ex-posure with the here presented PEAR OFR study?

o L564-566: The authors conclude: "Based on this work, different transformation pathways for RWC exhaust under photochemical conditions can be roughly outlined: the initial pathways consisting of functionalisation and condensation from gaseous precursors are followed by more particulate-phase-driven chemistry consisting of heterogeneous oxidation and fragmentation." The authors need to discuss their limitations of differentiating between gas-phase oxidation, heterogeneous chemistry and particle-phase-driven chemistry owing to their experimental set-up in the discussi-on of the results and the presentation of their conclusions.

o L569 onwards: this information is quite generic and could be omitted and replaced with more specific conclusions in my point of view; otherwise it does not add additional value to the manu-script.

Technical Comments:

o Formatting of units (e.g. L/min vs L min-1), figures (e.g. legends are sometimes to be found left, right or centered) and tables (e.g. horizontal lines in tables, as well as table dimensions), in parti-cular in the SI is inconsistent. While this is of course not critical with regards to the scientific quali-ty of the work, it would help the reader to follow the presented research work more easily and hence enjoy the results more.

Supporting Information

o All information provided in the SI should also be noted in the main text; all information provided in the SI should be described with the references such that the document can be read indepen-dently, etc.

• E.g., Table S1: please add reference for the OH constant used.

• E.g. Table S2.1: please add reference for the "OHR external" definition. Is CH4 negligible or why was it not included in the analysis?

REFERENCES

Bruns et al. 2015, DOI: 10.5194/amt-8-2315-2015 Czech et al. 2017, DOI: 10.1016/j.atmosenv.2017.03.040 Pieber et al. 2018, DOI: 10.1021/acs.est.8b04124

---

## Referee Comment (RC2) · Anonymous Referee #2 · 23 Jan 2020

Overview

The manuscript by Hartikainen et al investigates how gas- and particle-phase emissions from residential wood combustion vary with respect to combustion conditions (including stove type) and fuel. Additionally, the emissions are aged in a photochemical reactor to investigate how composition evolves with atmospheric age. Numerous analytical techniques are used allowing the authors to broadly characterize both gases and aerosols. Overall, the authors find that emissions depend on combustion conditions and that photochemical aging alters composition, generally by creating more oxi-

dized species. Emissions from residential wood combustion is an important and poorly understood contributor to air quality issues and the understanding of the influence of aging is poor. Thus, although this is a largely descriptive paper with few quantitative or testable conclusions, the experiments are of interest to the community. However, I have several major concerns that should be addressed prior to acceptance. My main critique is that the manuscript claims to investigate the aging that occurs over multiple days but there is no discussion about how the experimental conditions differ from the atmosphere nor is there discussion/consideration about how important reactions such as peroxy radical fate differ between the OFR and the real atmosphere.

Major Comments

1) The description and analysis of the OFR experiments is insufficient and requires substantial expansion. Interpreting the chemistry of OFRs is difficult and there needs to be careful consideration of the dilution effects, gas-phase peroxy radical fate, NO and NO2 mixing ratios, and potential for unwanted chemistry if the results are to be applied to the atmosphere. This is particularly true when making claims about multiple day aging timescales as is done here.

In terms of the description, details such as the mixing ratio of ozone and butanol should be included as should the residence time.

In terms of analysis, the authors need to more carefully consider the operation of the OFR, how this impacts the results, and the subsequent implications for atmospheric relevance. I list some specific questions below, but there needs to be a more general consideration of this chemistry.

For instance, in the atmosphere the emissions will experience dilution over the course of several days aging – how might dilution alter the implications of this work?

How representative is the peroxy radical chemistry (Peng et al., 2019) and how might this alter in particular the gas-phase measurements?

[Figure]

Is NO3 chemistry occurring in the reactor and if so, does it vary as a function of the OH exposure or across a given experiment? The formation of compounds such as nitroaromatics will depend on NO2. Is it possible that nitrophenols decreased with increased aging because the NO/NO2 chemistry was altered in the reactor and thus the formation of nitrophenols was altered (rather than nitrophenols being oxidized by the increased OH as is implied in the manuscript)? Overall, the OFR chemistry needs to be considered more thoroughly in order for meaningful conclusions to be drawn about how the emissions will be transformed in the atmosphere.

2) I find the manuscript difficult to read given the number of different variables explored and the number of analytical techniques used. While it is an advantage that multiple instruments measured the same thing, it is often not clear in the figures or the text which measurement or condition is being discussed. This makes it difficult for the reader to identify the main conclusions and findings. Clarification of the combustion/oxidation conditions and analytical instrumentation being discussed needs to be made more explicit throughout the text. For instance, in Fig. 6 are the values averaged over all the batches? I assume that Fig. S2 is FTIR measurements, but it would be useful to explicitly state.

Minor Comments

Sect 3.5.2 Did the authors consider performing PMF with the rBC peaks included? It would be interesting to see if the rBC peaks supported the PMF factor interpretation.

Line 494: The statement about diminished health effects is not well supported, particularly since it is followed with a statement that the heteroatom containing PACs may have negative health impacts. Without any measurements of for instance ROS generation, I think the more accurate statement is that the health effects would likely change (but no indication of better or worse).

Technical comments Why only consider m/z 40-180 for the PTR?

I think "oxygenated" rather than "oxidized" would be a better choice for describing the compounds measured in the unoxidized exhaust in order to avoid confusion (for instance in Fig. S3).

S3 and S4 are difficult to interpret since the x-axis and groupings are changed. It would be easier to compare if they were kept in the same format.

Line 65 and elsewhere, please clarify what is meant by "semi-VOCs"

Line 267 these aren't units of emissions

Line 287: Figure S14 referenced out of order. Other references may be out of order as well.

Line 319: What is meant by "external OH reactivity"?

Reference Peng, Z., Lee-Taylor, J., Orlando, J. J., Tyndall, G. S. and Jimenez, J. L.: Organic peroxy radical chemistry in oxidation flow reactors and environmental chambers and their atmospheric relevance, Atmospheric Chem. Phys., 19(2), 813–834, doi:https://doi.org/10.5194/acp-19-813-2019, 2019.

---

## Author Comment (AC1) · 10 Mar 2020

**Authors response to Anonymous Referee #1**       **Manuscript acp-2019-1078**

We thank the reviewer for this thorough feedback, which has been very useful in improving the composition and preciseness of the manuscript. In the following comments we provide point-by-point responses to the questions and comments by the referee. The replies to the comments are indicated as red text. The revised manuscript showing the changes made to the text is available in an additional comment in the manuscript discussion thread.

**GENERAL COMMENTS:**

This article adresses photochemical aging of residential wood combustion (RWC) emission from two stoves at the home-built PEAR oxidatation flow reactor, reaching OH exposures representative of aging of up to one week in the atmosphere. This is one of few studies adressing the topic of RWC aging in OFRs with a suite of state-of-the art mass spectrometric online and offline techniques and adds value for the scientific community.

In my view, the manuscript is of high scientific quality and presents relevant results from a comprehensive study. The language and structure, however, lack clarity and some information is redundant which makes the manuscript unnecessarily long and the core findings quite dilute. I suggest that this be improved to make the article more concise and easier to follow and understand for the readers. I believe this is feasible with major revisions of text and paragraph structure in order to allow more clarity on the main findings, conclusions and limitations of the study. Aside of specific comments below, re-formatting of units, table layouts, etc., into one consistent format and re-structuring paragraphs may help to achieve this. As a general comment, the results section would benefit if the authors would focus on summarizing their main findings first, and discuss the limitations of the study in a second stage with a focus on their implications of the obtained results, rather than first discussing experimental conditions/limitations as results followed by reporting the actual findings.

In summary, I have enjoyed seeing these results and am looking forward to see the revised manuscript published.

The entire text was revised to improve clarity of the manuscript, shorten the text, and improve language where necessary. Some information was moved to the supplementary information in order to better emphasize the core findings in the manuscript text. The tables and figures have been reformatted where necessary for better consistency in formats. We also added a paragraph in the Results & discussion section which summarizes the main findings as requested by the reviewer.

**SPECIFIC COMMENTS**:

**Overall:**

The main findings are dilute throughout the manuscript and difficult to grasp. Parts in the conclusion section remain vague too. I suggest to summarize the main goals and questions of the study as specific as possible at the end of the introduction and to state the core findings as specific as possible in the conclusions section.

The goals of the manuscript are now stated more clearly at the end of Introduction. We included a chapter in the beginning of the Results and discussion to give an outline of the contents and major findings in the result:

"This study comprehensively characterises the chemical properties of RWC exhaust at different atmospheric aging times by combining extensive information gathered from gas-phase and particulate phase chemical analyses. In this section, we first discuss the dynamic combustion conditions and the characteristics of primary emissions from logwood stoves utilized with different fuels, which define the starting point for the aging experiments. Next, the aging conditions in the PEAR OFR are evaluated in order to validate the atmospheric relevance of the results. Finally, we assess the changes in the gaseous and particulate OA during the aging process under a variety of different oxidant concentrations. The observations of changes both in bulk- and molecular level aerosol chemical composition demonstrate that the major transformation pathway of OA changes from initial gas phase functionalisation followed by condensation to the transformation of the particulate OA by heterogeneous oxidation reactions and fragmentation. The study shows a linear dependency between OH exposure and organic aerosol oxidation state. Furthermore, OH-exposure-dependencies of specific OA constituents, such as nitrophenols, carboxylic acids and PACs, are established."

We also thoroughly revised the Conclusion section, where needed.

While the manuscript title is focused primarily on the photochemical aging of the emissions, a substantial part of the manuscript discusses primary emissions (Fig 2+3), operating/experimental conditions (Fig 4+5) and only the second part of figures (Fig 6-11) and results actually addresses the aging and chemistry question. In my view, some of the initial figures (e.g. 4+5) could be moved to the SI along with their text in favor of clarity of the manuscript; the implications of the determined losses in Fig 4+5 should then be referred to discussing the actual results, i.e. for example, the authors might adress how do particle/vapor losses influence the observed decay of compounds as a function of OH exposure in their experiments. Alternatively, Figure S1, S3 and S4 could fit well with the main text, as they summarize the primary emissions composition, which is relevant for any following discussion of emissions aging. Further remarks to help clarity and focus are listed later on.

We consider presentation of the primary emission contents important for throughout understanding of the aging process and products, but the focus of this manuscript is indeed in the secondary products formed in the aging exhaust. Thus, as proposed by the reviewer, revised Figure S3 was implemented in the main text, replacing previous, more general Figure 3. Furthermore, information on the LVOC fate, including previous Figures 4 and 5, was transferred to Supplementary section S2.2.

**Abstract, Introduction & Methods:**

- L20/ Abstract: suggest to specify here which oxidation flow reactor is used in this study; suggest to use "PEAR OFR" throughout

Revised as suggested by the referee. The reactor is now referred to as "the PEAR OFR" throughout the manuscript.

- L21 vs L 28 and other occasions/ Abstract: The authors mix between "gaseous organic compounds" and "volatile organic compounds"; suggest they try to harmonize the language

We harmonized the language by using "organic gaseous compounds (OGC)" throughout the manuscript when the volatility of the compound/group was not mentioned.

- L27/ Abstract: specify based on which analysis the acetic fragmentation is investigated

The following sentence was added into the abstract:

"Aging led to an increase in acidic fragmentation products in both phases, as measured by the IDTD-GC-ToFMS for the particulate and PTR-ToF-MS for the gaseous phase."

- L30 / Abstract and other occasions: the authors use the more generic term "polycyclic aromatic compounds (PACs)" rather than polycyclic aromatic hydrocarbons (PAHs) which has been used in previous related work and is also mentioned in e.g., L75. I suggest to specify and/or define for non-specialists, why the authors refer to PACs in some cases but PAHs in others. This should also take into account the limitations of their instrumentation to effectively distinguish between PACs and PAHs. In my understanding, frequently only fragments of PACs can be detected with PTR-ToF-MS or AMS, however, they are assumed to be and referred to as PAHs in the manuscript.

We chose to use the term "PAC" due to it covering also substituted polycyclic aromatic compounds. It is also suitable in cases where the parent compound is not known. The use of the terms (PAC/PAH) in the manuscript was revised to be more uniform (e.g. "*X-substituted PAC*" instead of "*X-substituted PAH*").

Analysis of all PACs, including the substituted PAHs, was done in similar manner. Fortunately, fragmentation of the PACs is minor in both PTR-ToF-MS (Gueneron et al., 2015) and AMS (Herring et al., 2015). With the AMS, the PAC with m/z < 300 are seen as the original molecular ion, while for larger PAC only the possible fragments are observed in the spectra (< 5 % of AMS signal; Herring et al. 2015). PACs are known to give intense signatures in AMS for both single and double charged molecular ions (Herring et al., 2015; Dzepina et al., 2007) due to delocalization of the charge, which is an asset when considering the extent of possible fragmentation. For the HR-PAH analysis, the quantification by the P-MIP-method (Herring et al., 2015) is done by considering also the main fragmentation and isotopic patters of the PACs, which enables precise molecular identification likewise for substituted and for hydrocarbon PACs.

- L36 / Abstract: suggest to specify which scale is "fresh", "shortly aged", and "longterm aged" in the context of this study

During revision, this sentence was removed from the abstract, which now states, more broadly, that "The observed continuous transformation of OA composition throughout a broad range of OH exposures indicates that the entire atmospheric lifetime of the emission needs to be explored --"

- L54: Is there a difference between "fresh" and "unaged"? Otherwise, I suggest the authors rephrase to read "fresh, i.e. not atmospherically aged"

In this situation there is no difference. Revised.

- L60/61: The authors may include in this section of SOA-precursor discussion that removing aromatic hydrocarbons from wood burning emissions by use of e.g. catalytic converters can drastically reduce SOA formation, as recently presented by Pieber et al., 2018; also changing applicance operation and modifying combustion phases to conditions that emit less aromatic hydrocarbons might have this effect; Figure S3+S4 are valuable indicators in this context.

This is a good notion, and now also included in the Introduction:

"-- removal of these [aromatic] compounds either via improved combustion conditions or for example catalytic cleaning have been shown to be efficient in lowering the SOA potential of RWC emissions (Czech et al., 2017; Pieber et al., 2018). "

It is now also mentioned in the results section 3.2.1 to highlight the importance of aromatic compounds to SOA formation.

- L87: suggest to provide some references for previous RWC OFR studies in addition to the men-tioned smog chamber experiments; e.g. Bruns et al. 2015, Czeck et al. 2017, Pieber et al. 2018, etc.

References to previous OFR studies are now also included.

- L132: "In addition". . . "additional"; language is redundant. Suggest to remove one of the two.

We have removed the extra "additional" as suggested.

- L133-134: Ozone and butanol-d9 metrics in volumetric flows does not provide any information on concentration levels; could the authors in addition or instead specify the mixing ratios?

Ozone mixing ratios are available in Table 1 and now also included in the Methods-section, together with the butanol-d9 mixing ratios.

- L134: "formed" should probably read "forms"

Corrected.

- L135: "depended" should probably read "depends"

Corrected.

- L135: "H2O" should probably read "H2O vapor"

Corrected.

- L143: "OH concentration" should read "24-hour average global OH concentration"

Corrected.

- L146: was butanol-d9 mixing ratio included in this equation an what is ist relative contribution to the total OHR external?

Butanol-d9 was not included in the initial OHRext calculations. We thank the reviewer for pointing this out, as naturally the additional OH tracer consumes radicals as well. The $OHR_{ext}$ by butanol-d9 varied from 7 –17

s$^{-1}$, corresponding to $1 – 7$ % of total OHR$_{ext}$. Its importance was highest on the experiments with otherwise lower OHR$_{ext}$, and minor (1 %) on the experiments with e.g. lower DR (and higher OHRext).

OHR$_{ext}$ by butanol-d9 is now included in the discussion about OHR$_{ext}$ and in the consideration of the reaction pathways in the PEAR OFR, including e.g. Table S4 and Figs. S6, S10 and S11.

- L148: suggest the authors present the main results of OHR external analysis in one brief sentence in this paragraph, and mention its implications for the results and conclusions later on.

We now present the main OHRext results already in the Methods Section 2.2:

"Due to the differences in the emission concentrations during logwood combustion also the OHRext , and consequently OHexp, vary also within a batch (Fig. S9). Average OHR$_{ext}$ was in the range $130 – 1300$ s-1 with the highest contributions from CO, NO, and unsaturated hydrocarbons (Fig. S5)."

- L150ff: suggest the author present the main results of particle and LVOC loss estimates in 1-2 brief sentences in the main text here and move the remaining text on this topic from the results section to the SI, discussing it in the main text only as a limitation of the study including its implications for conclusions (as reasoned above). I suggest also to define what LVOCs refers to in the context of this article.

Discussion of LVOC fate from chapter 3.3.1. was divided into main text Section 2.2. and SI Section S2.2. as suggested. We rephrased the whole paragraph, which now also better defines the meaning of "LVOC" in this context, i.e.: "The fate of the gaseous organic compounds capable of irreversible condensation under the present experimental conditions, i.e., low-volatility organic compounds (LVOC), was estimated based on Palm et al. (2016)".

- Line 160/161: I suggest to keep all information in one paragraph, rather than starting a new para-graph here.

Corrected as suggested.

- L167: suggest to define semi-VOCs in this context.

Section was revised to note that majority of the compounds can classified as semi- or intermediately volatile (saturation mass concentrations between $0.3 - 3 \times 10^6$ ), to highlight that these compounds may exist in both phases under the experimental conditions.

- L168: "isotope labelled" should probably read "isotopically labelled"

Corrected.

- L173: suggest to replace "after the PEAR" with "after the PEAR, i.e. at its outlet"

Revised to " -- were monitored at the outlet of the PEAR OFR"

- L187: the author refer to this instrument as "AMS" throughout the manuscript; I suggest to chan-ge this to SP-HR-ToF-AMS on all occasions, starting from the abstract, as is done for all other instrumentation (e.g. PTR-ToF-MS).

AMS is now referred to as SP-HR-TOF-AMS throughout the manuscript.

- L189: Why did the authors choose to use the "Improved-Ambient method" for this laboratory dataset?

The commonly used "Aiken-Ambient" method is systematically biased low (Aiken et al., 2008), with larger biases observed for alcohols and simple diacids. The Aiken-Ambient method underestimates the $CO^+$ and especially $H_2O^+$ produced many oxidized species. The Improved-Ambient method uses specific ion fragments as markers to correct for molecular functionality-dependent systematic biases and reproduces known O:C and (H : C) ratios of individual oxidized standards within 28 % (13 %) of the known molecular values (Canagaratna et al., 2015).

- L200: Suggest to compare also with Bruns et al., 2015 who have used this method beforehand for an intercomparison of spectra from OFR and smog chambers including RWC.

Bruns et al. (2015) applied the PMF as a tool for separation of POA/SOA components and created a two-factor solution by first finding the mass spectra for POA, and then assuming the rest as SOA (or aged-POA). In other words, spectra for SOA is not given nor were different oxidation pathways segregated. One important difference between PMF factors by Bruns et al. (2015) and those present here, is that in Bruns et al. the exhaust from the batch was first mixed in a chamber before aging in a PAM OFR, while in this manuscript the measurement of the exhaust was dynamic, and the factors are also tied to the combustion periods of batchwise combustion with the POA factors being separated by the combustion phases.

References to previous RWC studies utilizing similar factorization methods are now included in Section 2.5:

"Similar method have been used previously for example for the assessment of RWC generated POA and SOA in a chamber (Bruns et al., 2015a; Tiitta et al., 2016) and in an OFR (Bruns et al., 2015a) or for time-resolved analysis of RWC OA emission constituents (Elsasser et al., 2013; Czech et al., 2016). "

- L205: Suggest to replace "residential wood combustion" with RWC

Replaced as suggested.

- L209 (from L186 onwards): as noted in the next comments; please add important information on AMS data analysis in the main text.

Information of the AMS operating conditions was moved to the Material and methods Section S2.5.

Some text is repetitive or split into different locations (partially found in main text, partially in duplicate SI), while other text is grouped together into subchapters which are not entirely logical. I suggest to make the overall language more concise and shorten the text, while keeping all relevant information in the main text. E.g. some suggestions:

- L170 (main text): I suggest to relabel this as "Online aerosol particle and gas-phase measurements" to avoid ambiguity." It may be beneficial to split the subsection 2.4 into three sections for each instrument, and then make further separat paragraphs for any AMS in-formation aggregated as a) experimental/calibration/raw data correction, b) PMF analysis, c) HR fitting or PAH fitting.

We split the section 2.4 into two sections, namely, 2.4 for the gas phase measurements, 2.5. for the particulate-phase. AMS section now contains three paragraphs which are related to (1) data correction, (2) PMF-analysis, and (3) HR-PAH analysis.

- L186 onwards (main text): relevant AMS operating conditions (e.g. size-cut off) and data corrections (RIE=1.4 for OA, CO2+-interference correction, CO2 gasphase correction) should be mentioned in the main text; I suggest to move information from SI (Section S5) entirely to the main text, but shorten the paragraph by avoiding redundencies (e.g. "The AMS data was analysed using the standard analysis tools SQUIRREL v1.62A and PIKA v1.22D adapted in Igor Pro 8 (Wavemetrics)." is currently stated double.

Information of the AMS operating conditions was moved to the main text (Section 2.5) and the repetitive parts removed or rephrased; see specific comments for rephrased sections below.

- With regards to the RIE=1.4, please mention that this is valid for OA.

For biomass burning emissions, RIE of 1.4 agrees with the OA mass concentrations with stated ±38 % uncertainty of AMS (Bahreini et el., 2009; Xu et al. 2018). This is now stated in Section 2.5 discussing AMS analysis.

- With regards to the CO2+-interference correction, please mention the magnitude of interference and potential impacts on the determined O/C ratios.

A $CO_2$-AMS interference calibration value of 0.3 % (a=0.003) of the $NO_3$ concentration was determined by NO3NH4 calibration and corrected via the fragmentation table according to Pieber et al. (2016). Without correction, maximum effect on O:C value was 5 %, with typical values lying under 1 %. This is now stated in the Section 2.5 on AMS analyses.

- What was the level of inorganic nitrate to OA in the primary and secondary emissions?

The concentrations of inorganic $NO_3$ in the emissions are available in Table S7. The ratio of $NO_3$-to-OA varied between 0.02 (masonry heater, age 1.5 eqv.d) and 1.1 (chimney stove, age 0.6 eqv.d), but cannot be tied to atmospheric age. As stated in the previous answer, the effect of nitrate to the AMS analysis was small.

- With regards to CO2 gas phase-correction, please mention whether this was done by standard measurements of particle-filtered air during the experiment or external calibration with gaseous CO2.

The time-dependent gas-phase $CO_2^+$ subtraction was applied. It was conducted using the online HEPA filter measurement technique of gas-phase $CO_2$ for corrections. This information was added to the Material and methods Section 2.5 on AMS analysis.

- L190 onwards (main text): I suggest to make a new paragraph with all information relevant to PMF, see comments above.

The information considering PMF was separated into its own paragraph, as suggested by the reviewer.

**Results:**

- L210: Suggest to relabel as "results and discussion", given the separate section entitled "conclusions" in L540

Corrected.

- Section 3.1.: Is there any information on the temperatures during these different combustion phases available?

Unfortunately, no measured temperature information is available from the current study. The time-dependent temperature patterns during batchwise logwood combustion and their effects on emission characteristics have been established earlier by e.g. Czech et al., (2016) and Kortelainen et al., (2018). Section 3.1 was modified to not give an impression that we have exact measured temperature data available from these experiments.

- L228: "constantly" should probably read "continuously"; "from the diluted flue gas" should pro-bably read "in the diluted flue gas".

Corrected.

- L232: "primary aerosol" is certainly technically correct if referring to aerosol as particles plus their surrounding gas, however, it may quickly become confusing as often "aerosol" is used when refer-ring to the particle phase only. I suggest to replace "primary aerosol" with "primary emissions" here.

This sentence was revised to "-- OGC in the primary emissions --" to highlight that here we discuss the gaseous phase of the primary emissions.

- L233: suggest to specify the value of protonation efficiency (proton affinity) here

Mention of the exact proton affinity of water (691 kJ mol$^{-1}$) that sets apart the detectable compounds is now included.

o L230: suggest to state the limitations of both, FTIR and PTR-ToF-MS, here briefly, e.g. for PTR-ToF-MS, rather than point out alkanes specifically, it should be noted that any molecules with proton affinity lower than that of water (in H3O+ mode) can not be protonated and hence detected; further, I suggest to mention based on which criteria 127 molecular formulas were identified and how many ions remained unidentified. For the FTIR, it would be informative to give a brief reasoning why only 27 compounds can be detected and quantified, rather than only refer to the table in SI.

The sentence about PTR limitations is now revised, including e.g. the limiting protonation affinity, as suggested. The identification of compounds by PTR-MS was done based on the high-resolution m/z and previously reported compounds found from RWC as in previous work (Hartikainen et al. 2018) as stated on Section 2.4. Molecular formula could be allocated to majority of the compounds, although precise identification of higher m/z compounds is innately not possible.

The organic compounds measured with the FTIR were limited to those calibrated to the particular instrument used in this study. The sentence describing the measured components in the manuscript Section 3.2.1 now states that "-- FTIR was calibrated for 27 typical combustion-derived compounds --"

- L233: fragmentation does not necessarily limit the quantification but rather the identification; suggest to rephrase

True. The sentence is rephrased.

- L236: suggest to introduce the later used abbreviation "ArHC" here at its first occasion

Introduction of the abbreviation is now included already in Section 3.2.1.

- L243: suggest to replace "significant" by "statistically significant" here

Done.

- L259: ArHC are also discussed earlier, but the abbreviation is only introduced in L259; suggest to revise and use and define ArHC earlier on

ArHC is now defined already in the Section 3.2.1 when discussing primary OGC emissions.

- L265: Suggest to add that previous studies (e.g. Pieber et al. 2018), demonstrated that removing ArHC from the emissions mix substantially reduces their SOA formation.

A notion that " removal of ArHC from the flue gas either by improving combustion conditions or using e.g. catalytic cleaning has been noted to decrease also the resulting SOA formation (Bruns et al., 2016; Hartikainen et al., 2018; Pieber et al., 2018). " were added to Section 3.2.1, as suggested.

- Figure 7 and 8: where do O/C and H/C ratios of other wood burning OFR studies fall in compari-son to the obtained results (e.g. add data from Bruns et al., 2015, Czech et al. 2017, Pieber et al., 2018, and similar data from other research groups as available).

The OSc of highly aged particulate OA observed here is lower than previously measured for RWC exhaust with a PAM OFR (Bruns et al., 2015, Pieber et al., 2018), where the exhaust from batchwise combustion was mixed prior to aging in the PAM. While this approach is reasonable as the exhaust exists as a mixture also in the atmosphere, the resulting O:C and H:C ratios (see the van Krevelen-diagram below (Fig. 6 in the revised manuscript)) indicate worse agreement with the ambient range when compared to compositions obtained by the PEAR OFR (presented in this work) or previously measured in a chamber (Tiitta et al., 2016). This can also be due to differences in the combustion technologies and protocols or differences in the aging conditions.

Comparison of the O:C and H:C ratios of the particulate OA to previous studies in the Section 3.5.1 is now expanded. The results of aging of RWC exhaust by PAM OFR (by Bruns et al., 2015 and Pieber et al., 2018) are included also in the van Krevelen diagram (previously Fig. 8, now Fig. 6). Values from Czech et al. (2017) were not included in the van Krevelen diagram due to different combustion source (pellet boiler with notably higher initial oxidation states) but are included in the discussion on $OS_C$ on Section 3.5.1.

[Figure]

**Slopes:** Masonry heater, S-5% : -0.49, $R^2$ : 0.97; S-22% : -0.46, $R^2$ : 0.98
Chimney stove, B-17% : -0.34, $R^2$ : 0.66; S-22% : -0.17, $R^2$ : 0.49

Regarding the gaseous phase (Previous Fig. 7, now Fig. 5), OFR studies reporting gaseous compounds from RWC extensively are scarce and direct comparison of O:C and H:C ratios is not possible due to e.g. differences in the analyzed and reported compounds. Previously, Bruns et al. (2017) have reported emission factors for primary and secondary exhaust for 65 compounds measured with a PTR-MS. Although these compounds differ slightly from those considered here, the main compounds (aromatics and carbonyls with highest concentrations) are included. The changes during aging reported by Bruns et al. (2017) follow the phenomena observed here: both O:C and H:C ratios increase by photochemical exposure. The ΔH:C/ΔO:C during the five chamber experiments in Bruns et al. (2017) are in the range of 0.34 - 0.81 at final exposure of 1.9-2.3 eqv.d, which is similar to the changes observed here. This is now noted also in Section 3.4.

- L371: Replace "consequent" with "subsequent"

Corrected.

**Conclusions**

- L556: "Notably, small, acidic" should probably read "Notably, small acidic" (i.e. without comma)

Corrected.

- L565: given that the presented manuscript discusses OFR-experiments, I suggest to cite and discuss also other OFR studies with RWC rather than only smog chamber experiments; e.g. Czech et al. 2017, Bruns et al. 2015, Pieber et al. 2018: how do those PAM OFR studies compare in OH exposure with the here presented PEAR OFR study?

The photochemical exposures were in the ranges 0.5-2.5 eqv.d in Pieber et al. (2018) and ~18 eqv.h in Czech et al. (2017) who discuss pellet burning. To our knowledge, highest OH exposures for RWC exhaust in a PAM have been reported by Bruns et al. (2015), where the exposure was 2.5-10 eqv.d for the flaming phase of RWC. The conclusions section was overall revised, including rephrasing of this paragraph and addition of references to OFR studies, which are now also otherwise better noted in this work.

L564-566: The authors conclude: "Based on this work, different transformation pathways for RWC exhaust under photochemical conditions can be roughly outlined: the initial pathways consisting of functionalisation and condensation from gaseous precursors are followed by more particulate-phase-driven chemistry consisting of heterogeneous oxidation and fragmentation." The authors need to discuss their limitations of differentiating between gas-phase oxidation, heterogeneous chemistry and particle phase- driven chemistry owing to their experimental set-up in the discussion of the results and the presentation of their conclusions.

Limitations in differentiating between gas-phase oxidation, heterogeneous chemistry and particle phase -driven chemistry certainly exist. In addition, it should be noted that the different oxidation processes are likely overlapping. The sentence in question was aimed to simply give a rough outline of the major oxidation mechanisms, and is based on the following facts observed in this study: (1) for RWC-exhaust, short OH-exposures are sufficient to functionalize gaseous precursors and lead to their condensation into particulate phase, which consequently dominates the overall OA transformation until the SOA precursors have been depleted; (2) continuing aging in the presence of OH-radicals leads to further oxidation of particulate organic aerosol, which is likely explained by heterogenous reactions between gas-phase oxidants and particles. However, related to the second fact it is also possible that particulate phase oxidation occurs via evaporation and homogeneous gas-phase oxidation followed by recondensation, as discussed by for example Tiitta et al. (2016). Third, it should not be omitted that in an OFR utilized with high OH-radical concentrations, the gaseous phase precursors may receive higher oxidation state before condensation, e.g. due to several fast functionalization reactions in the gas phase, which would be a topic for further studies. Nevertheless, the comparison of similar OH-exposure in a smog chamber (low OH-radical concentrations) and  the PEAR OFR (high OH-radical concentrations) give fairly similar OA oxidation states, as stated e.g. in Section 3.5.1. and shown in Figure 6 (van Krevelen -diagram). This indicates that the utilized high OH-concentrations do not lead to any severe artefacts in terms of OA composition.

We have revised the text in the Conclusions section to better reflect on the limitations and the observations on which the conclusions are based upon.

- L569 onwards: this information is quite generic and could be omitted and replaced with more specific conclusions in my point of view; otherwise it does not add additional value to the manuscript.

This last, concluding paragraph considers the potential uses of these results, especially the importance of the consideration of different atmospheric aging stages for OA. The aim of the section in question is to emphasize the need for similar consideration of also other OA sources as a point toward future studies, pointing also towards the potential importance of aging towards both environmental and health related effects.

We have revised this final section of the Conclusions section, which now reads:

"--this study highlights the importance of also higher exposure levels towards chemical transformation of OA. Due to the potentially long atmospheric lifetimes of OA, long-term aging is also important to consider in large-scale atmospheric models, which typically estimate SOA formation and characteristics based on short-term aging experiments. The

consideration of only the first stage of gas-phase functionalization and condensation may lead to underestimated oxygenation of the long-transported OA, while specific compound groups, such as nitrophenols or substituted-PACs, can be overestimated. In general, the potential health and climate effects of aerosols are to a large extent determined by their composition, which depends on their sources and the levels of atmospheric aging. Thus, the characterisation of aerosol emissions from different sources and their atmospheric transformation at different exposure levels would be crucial for assessment of the overall environmental effects of ambient air pollution."

**Technical Comments**

- Formatting of units (e.g. L/min vs L min-1), figures (e.g. legends are sometimes to be found left, right or centered) and tables (e.g. horizontal lines in tables, as well as table dimensions), in particular in the SI is inconsistent. While this is of course not critical with regards to the scientific quality of the work, it would help the reader to follow the presented research work more easily and hence enjoy the results more.

The notations for units were corrected. Tables were made more consistent with e.g. no horizontal lines, and the figures were revised for better coherency, with similar formatting of figures of similar type (e.g. scatterplots Figs. 9-10 or timeseries in Figs. S6 and S10-S11.

**Supporting Information**

- All information provided in the SI should also be noted in the main text; all information provided in the SI should be described with the references such that the document can be read independently, etc.

All portions of SI are noted in the main text and clarification and references included in the text where needed.

- E.g., Table S1: please add reference for the OH constant used.

Reference included.

- E.g. Table S2.1: please add reference for the "OHR external" definition. Is CH4 negligible or why was it not included in the analysis?

Reference to the $OHR_{ext}$ is now included on Table S2.1. Methane ($CH_4$) is included in the analysis (6[th] compound from the bottom of the Figure S2), although its share in the total $OHR_{ext}$ is minor ($< 0.025\,\%$) due to its low OH reactivity.

REFERENCES

Bruns et al. 2015, DOI: 10.5194/amt-8-2315-2015 Czech et al. 2017, DOI:

10.1016/j.atmosenv.2017.03.040 Pieber et al. 2018, DOI: 10.1021/acs.est.8b04124

**REFERENCES**

Aiken, A. C., DeCarlo, P. F., Kroll, J. H., Worsnop, D. R., Huff-man, J. A., Docherty, K. S., Ulbrich, I. M., Mohr, C., Kimmel, J. R., Sueper, D., Sun, Y., Zhang, Q., Trimborn, A.,Northway, M., Ziemann, P. J., Canagaratna, M. R., Onasch,T. B., Alfarra, M. R., Prevot, A. S. H., Dommen, J., Duplissy,J., Metzger, A., Baltensperger, U., and Jimenez, J. L. O/C and OM/OC Ratios of Primary, Secondary, and Ambi-ent Organic

Aerosols with High-Resolution Time-of-Flight Aerosol Mass Spectrometry. Environ. Sci. Technol., 42, 4478–4485, doi:10.1021/es703009q, 2008.

Bahreini, R., Ervens, B., Middlebrook, A. M., Warneke, C., de Gouw, J. A., DeCarlo, P. F., Jimenez, J. L., Brock, C. A., Neuman, J. A., Ryerson, T. B., Stark, H., Atlas, E., Brioude, J., Fried, A., Holloway, J. S., Peischl, J., Richter, D., Walega, J., Weibring, P., Wollny, A. G., and Fehsenfeld, F. C. Organic Aerosol Formation in Urban and Industrial Plumes near Houston and Dallas, Texas. J. Geophys. Res. Atmos, 114, doi:10.1029/2008JD011493, 2009.Canagaratna et al., 2015.

Bruns, E. A., El Haddad, I., Keller, A., Klein, F., Kumar, N. K., Pieber, S. M., Corbin, J. C., Slowik, J. G., Brune, W. H. and Baltensperger, U. and Prévôt, A. S. H.: Inter-comparison of laboratory smog chamber and flow reactor systems on organic aerosol yield and composition, Atmos. Meas. Tech, 8, 2315-2332, doi:10.5194/amt-8-2315-2015, 2015.

Bruns, E. A., Slowik, J. G., El Haddad, I., Kilic, D., Klein, F., Dommen, J., Temime-Roussel, B., Marchand, N., Baltensperger, U. and Prévôt, A S H: Characterization of gas-phase organics using proton transfer reaction time-of-flight mass spectrometry: fresh and aged residential wood combustion emissions, Atmos. Chem. Phys., 17, 705-720, doi:10.5194/acp-17-705-2017, 2017.

Czech, H., Sippula, O., Kortelainen, M., Tissari, J., Radischat, C., Passig, J., Streibel, T., Jokiniemi, J. and Zimmermann, R.: On-line analysis of organic emissions from residential wood combustion with single-photon ionisation time-of-flight mass spectrometry (SPI-TOFMS), Fuel, 177, 334-342, doi:10.1016/j.fuel.2016.03.036, 2016.

Dzepina, K., Arey, J., Marr, L. C., Worsnop, D. R., Salcedo, D., Zhang, Q., Onasch, T. B., Molina, L. T., Molina, M. J. and Jimenez, J. L.: Detection of particle-phase polycyclic aromatic hydrocarbons in Mexico City using an aerosol mass spectrometer, International Journal of Mass Spectrometry, 263, 152-170, doi:doi:10.1016/j.ijms.2007.01.010, 2007.

Hartikainen, A., Yli-Pirilä, P., Tiitta, P., Leskinen, A., Kortelainen, M., Orasche, J., Schnelle-Kreis, J., Lehtinen, K. E., Zimmermann, R., Jokiniemi, J. and Sippula O.: Volatile organic compounds from logwood combustion: emissions and transformation under dark and photochemical aging conditions in a smog chamber, Environ. Sci. Technol., 52, 4979-4988, doi:10.1021/acs.est.7b06269, 2018.

Kortelainen, M., Jokiniemi, J., Tiitta, P., Tissari, J., Lamberg, H., Leskinen, J., Rodriguez, J. G., Koponen, H., Antikainen, S., Nuutinen, I., Zimmermann, R. and Sippula, O..: Time-resolved chemical composition of small-scale batch combustion emissions from various wood species, Fuel, 233, 224-236, doi:10.1016/j.fuel.2018.06.056, 2018.

Pieber, S. M., El Haddad, I., Slowik, J. G., Canagaratna, M. R., Jayne, J. T., Platt, S. M., Bozzetti, C., Daellenbach, K. R., Fröhlich, R., Vlachou, A., Klein F., Dommen, J., Miljevic, B., Jiménez J. L., Worsnop D. R., Baltensperger, U. and Prévôt A. S. H.: Inorganic salt interference on CO2 in aerodyne AMS and ACSM organic aerosol composition studies, Environ. Sci. Technol., 50, 10494-10503, doi:10.1021/acs.est.6b01035, 2016.

Pieber, S. M., Kambolis, A., Ferri, D., Bhattu, D., Bruns, E. A., Elsener, M., Kröcher, O., Prévôt, A. S. and Baltensperger, U.: Mitigation of Secondary Organic Aerosol Formation from Log Wood Burning Emissions by Catalytic Removal of Aromatic Hydrocarbons, Environ. Sci. Technol., 52, 13381-13390, doi:10.1021/acs.est.8b04124, 2018.

Xu, W., Lambe, A., Silva, P., Hu, W., Onasch, T., Williams, L., Croteau, P., Zhang, X., Renbaum-Wolff, L., Fortner, E., Jimenez, J. L., Jayne, J., Worsnop, D. and Canagaratna, M.: Laboratory evaluation of species-dependent relative ionization efficiencies in the Aerodyne Aerosol Mass Spectrometer, Aerosol Sci. Tech., 52, 626-641, doi:10.1080/02786826.2018.1439570, 2018.

---

## Author Comment (AC2) · 10 Mar 2020

**Response to Anonymous Referee #2 Manuscript acp-2019-1078**

The authors thank the reviewer for the comments, which improved especially the description of the atmospheric relevance of the results through the revision of the description of the aging conditions during the experiments. In the following comments we provide point-by-point responses to the questions and comments made by the referee. The replies to the comments are indicated as red text. The revised manuscript showing the changes made to the text is available in an additional comment in the manuscript discussion thread.

**Overview**

The manuscript by Hartikainen et al investigates how gas- and particle-phase emissions from residential wood combustion vary with respect to combustion conditions (including stove type) and fuel. Additionally, the emissions are aged in a photochemical reactor to investigate how composition evolves with atmospheric age. Numerous analytical techniques are used allowing the authors to broadly characterize both gases and aerosols. Overall, the authors find that emissions depend on combustion conditions and that photochemical aging alters composition, generally by creating more oxidized species. Emissions from residential wood combustion is an important and poorly understood contributor to air quality issues and the understanding of the influence of aging is poor. Thus, although this is a largely descriptive paper with few quantitative or testable conclusions, the experiments are of interest to the community. However, I have several major concerns that should be addressed prior to acceptance. My main critique is that the manuscript claims to investigate the aging that occurs over multiple days but there is no discussion about how the experimental conditions differ from the atmosphere nor is there discussion/consideration about how important reactions such as peroxy radical fate differ between the OFR and the real atmosphere.

**Major Comments**

1) The description and analysis of the OFR experiments is insufficient and requires substantial expansion. Interpreting the chemistry of OFRs is difficult and there needs to be careful consideration of the dilution effects, gas-phase peroxy radical fate, NO and NO2 mixing ratios, and potential for unwanted chemistry if the results are to be applied to the atmosphere. This is particularly true when making claims about multiple day aging timescales as is done here.

In terms of the description, details such as the mixing ratio of ozone and butanol should be included as should the residence time. In terms of analysis, the authors need to more carefully consider the operation of the OFR, how this impacts the results, and the subsequent implications for atmospheric relevance. I list some specific questions below, but there needs to be a more general consideration of this chemistry.

Description of the aging conditions and reactor use is naturally crucial for an OFR study, and consideration of the questions pointed out by the reviewer clearly improves the credibility of this assessment in this manuscript. In the following sections we provide point-to-point answers to the specific questions presented by dividing the comments by topic.

Mixing ratios of $O_3$ (2-11 ppm) are included in Table 1 and are now indicated more pronouncedly in the text of Material and methods Section 2.2 together with the initial mixing ratio of butanol-d9, which was 80-200 ppb. Residence time (139 s) is now also stated clearly in the Section 2.2.

For instance, in the atmosphere the emissions will experience dilution over the course of several days aging – how might dilution alter the implications of this work?

Dilution has important impacts to the gas-particle partitioning of organic aerosol. Under higher dilutions, semivolatile compounds measured in this work would have had higher concentrations in the gas phase and, in turn, lower concentrations in the particulate phase from where they were assessed with the IDTD-GCMS. During the lifetime of the emissions from the emission source until long-range transported smoke, a high range of different dilution ratios exist. For practical reasons (e.g. limited measurement time), we could perform the experiment only by using a certain range of dilution ratios. The usage of very high dilution ratios, which would be most representative to atmosphere, would be possible to in the OFR, but it would hamper the possibilities for comprehensive chemical and physical analyses of aerosols, due to sensitivity limits of the instruments. Thus, we used lower dilution in the PEAR than what normally occurs in atmosphere but have the advantage of better quality in the chemical analysis results. This is now mentioned also in the text (Section 3.3).

How representative is the peroxy radical chemistry (Peng et al., 2019) and how might this alter in particular the gas-phase measurements?

Due to the high OH and $HO_2$ concentrations in the PEAR OFR, the shortened lifetime of $RO_2$ decreases its isomerization compared to atmospheric conditions (Peng et al., 2019). Further, the HO2-to-OH ratio (roughly approximating in the range of ~10-50:1) is lower than in atmosphere (~100:1). Consequently, the importance of $RO_2$+OH reactions is enhanced. The $RO_2$ fate in the PEAR OFR was roughly estimated using the $RO_2$ fate estimator by Peng et al. (2019). The estimation of the $RO_2$ fates during the first minute, during which majority of the OGC have reacted, is given below for two different experiments: first with high OH exposure (Masonry heater Exp. 2), second with high $OHR_{ext}$ (Chimney stove Exp. 5).

[Figure]

Discussion of the peroxy radical chemistry is now included in Section 3.3 discussing OFR chemistry and noted also in the discussion of VOC transformation, as fragmentation in the PEAR OFR may be enhanced due to the different $RO_2$ chemistry than in atmosphere or in chamber experiments.

Is $NO_3$ chemistry occurring in the reactor and if so, does it vary as a function of the OH exposure or across a given experiment?

We assume $NO_3$ to be negligible due to its slow formation rate and fast photolysis in the PEAR. This is supported by the low AMS $NO$-to-$NO_2$ ratio, based on which all particulate nitrate was inorganic.

The formation of compounds such as nitroaromatics will depend on NO2. Is it possible that nitrophenols decreased with increased aging because the NO/NO2 chemistry was altered in the reactor and thus the formation of nitrophenols was altered (rather than nitrophenols being oxidized by the increased OH as is implied in the manuscript)?

Although the ratio of $NO$-to-$NO_2$ prior to aging was in the range of ~10-40:1, NO is rapidly oxidized to $NO_2$ in the PEAR OFR and the amount of NO measured downstream the PEAR OFR is negligible (<0.02 ppb) in all experiments. Similar $NO$-to-$NO_2$ conversation takes place also in environmental chamber, where formation of nitrophenols has been observed previously (Hartikainen et al., 2018). Furthermore, exposure level was not observed to affect the observed $NO_2$ levels.

Although we cannot explicitly state the reason for nitrophenol decay in the scope of this manuscript, we find it likely that the after the rapid conversion of the precursor compounds (i.e., phenols) has taken place at the first stages of aging, the change in nitrophenol concentrations in the PEAR OFR is governed by nitrophenol-OH reactions and photolysis (Hems and Abbatt, 2018), which are both enhanced in the high-aging experiments due to the high photon fluxes and OH concentrations.

Overall, the OFR chemistry needs to be considered more thoroughly in order for meaningful conclusions to be drawn about how the emissions will be transformed in the atmosphere.

We hope that the above-mentioned improvements on the explanations of the chemistry in the PEAR OFR and the overall revision of the manuscript text now provide satisfactory information for validation of the used method and comparison against atmospheric conditions.

2) I find the manuscript difficult to read given the number of different variables explored and the number of analytical techniques used. While it is an advantage that multiple instruments measured the same thing, it is often not clear in the figures or the text which measurement or condition is being discussed. This makes it difficult for the reader to identify the main conclusions and findings. Clarification of the combustion/oxidation conditions and analytical instrumentation being discussed needs to be made more explicit throughout the text. For instance, in Fig. 6 are the values averaged over all the batches? I assume that Fig. S2 is FTIR measurements, but it would be useful to explicitly state.

In Figure 6 (now Fig. 4) the values are indeed averaged over all the batches per experiments and Fig. S2 is based on the FTIR measurements. Clarification of the conditions and analytical instrumentation employed for each finding was added when necessary, for example in Sections 3.5.4 and 3.5.5 and in the supplementary figures (Figs. S2, S5-S8, S16-S17 and S19).

**Minor Comments**

Sect 3.5.2 Did the authors consider performing PMF with the rBC peaks included? It would be interesting to see if the rBC peaks supported the PMF factor interpretation.

We thank referee for the valuable idea of performing PMF analyses with the rBC peaks included. However, it falls outside of the main focus of this paper, which is the dependence of organic aerosol composition on OH exposure.

We performed a test run with PMF applying SP-AMS data using refractory carbon clusters. Two-factor solution showed that timeseries of the rBC-dominated factor correlated best with the timeseries of flaming-BBOA factor (see figure below). This observation supports the PMF interpretation since it is well-known that the refractory black carbon is formed mainly during the flaming phase (Kortelainen et al., 2018; Nielsen et al., 2017).

[Figure]

In future work we would also like to connect refractory black carbon with the high-resolution NR-OA to analyze rBC-associated coating of OA.

Line 494: The statement about diminished health effects is not well supported, particularly since it is followed with a statement that the heteroatom containing PACs may have negative health impacts. Without any measurements of for instance ROS generation, I think the more accurate statement is that the health effects would likely change (but no indication of better or worse).

True. This sentence in Section 3.5.3 was revised, and now reads:

"The change in the PACs also alters the potential health effects of the exhaust: although the total PAC concentration decayed, the simultaneous formation of oxy- and nitro-PAC derivatives known to be detrimental to health was observed."

**Technical comments**

Why only consider m/z 40-180 for the PTR?

According to our experience, the compounds reliably measurable from RWC exhaust with PTR fall within this range, which was stated mainly to specify the observable range to the reader. The sentences in Section 3.2.1 and 3.4. describing the use of PTR-MS are now rephrased ("-- for OGC in the primary emissions were identified in the m/z range of 40–180 --" and "aging may also lead to the growth of compounds outside the observable mass range", respectively).

I think "oxygenated" rather than "oxidized" would be a better choice for describing the compounds measured in the unoxidized exhaust in order to avoid confusion (for instance in Fig. S3).

This is true. We also used "oxygenated" instead of "oxidized" in the text and for example in the previous Fig. 3 and Table S1. The description is now unified and "oxygenated" used also in e.g. Fig. S2 and the previous Fig. S3 (now revised and replacing Fig. 3 in the main text).

S3 and S4 are difficult to interpret since the x-axis and groupings are changed. It would be easier to compare if they were kept in the same format.

Figure S3 was revised to share the format of Fig. S4 (now S3) and moved to the main text where it now replaces the previous Fig. 3.

Line 65 and elsewhere, please clarify what is meant by "semi-VOCs"

Rephrased to more general term, organic gaseous compound (OGC).

Line 267 these aren't units of emissions

We revised the terminology to "concentrations" in this section (3.2.2) considering the primary emissions. The concentrations in the secondary exhaust are normalized to 13 % flue gas excess oxygen, which is a common procedure to present emissions from logwood fired stoves. The fact that the emission concentration is normalized to a certain oxygen level means that the values corrected for the changing air-to-fuel ratios and are therefore directly proportional to emission factors as #/fuel energy content or #/consumed fuel mass. The emission factor calculation procedures are presented more thoroughly by e.g. Reda et al., (2015, supplement). We have now also stated clearly in e.g. figures which concentrations are normalized to 13% flue gas excess oxygen.

Line 287: Figure S14 referenced out of order. Other references may be out of order as well.

Due to the length of the supplementary information, it is structured by subject matter. In other words, all material related to each subject are all grouped under the respective title, rather than included in the order of first appearance. We also decided to keep figures with the same structure/topic together, which is the case with Fig. S14 (now Fig. S15), where thermal-optical EC and OC are compared to AMS rBC and OA, respectively.

In some cases, this also leads to references in the main text to be out of order. With this, we aim to help the reader in finding the information related to each subject.

Line 319: What is meant by "external OH reactivity"?

External OH reactivity refers to the OH reactivity of the gases in the sample that is input PEAR OFR (Li et al., 2015). It is defined in the Material and methods section 2.2.

Reference Peng, Z., Lee-Taylor, J., Orlando, J. J., Tyndall, G. S. and Jimenez, J. L.: Organic peroxy radical chemistry in oxidation flow reactors and environmental chambers  and their atmospheric relevance, Atmospheric Chem. Phys., 19(2), 813–834, doi:https://doi.org/10.5194/acp-19-813-2019, 2019.

**REFERENCES**

Hartikainen, A., Yli-Pirilä, P., Tiitta, P., Leskinen, A., Kortelainen, M., Orasche, J., Schnelle-Kreis, J., Lehtinen, K. E., Zimmermann, R., Jokiniemi, J. and Sippula O.: Volatile organic compounds from logwood combustion: emissions and transformation under dark and photochemical aging conditions in a smog chamber, Environ. Sci. Technol., 52, 4979-4988, doi:10.1021/acs.est.7b06269, 2018.

Hems, R. F. and Abbatt, J. P.: Aqueous phase photo-oxidation of brown carbon nitrophenols: reaction kinetics, mechanism, and evolution of light absorption, ACS Earth Space Chem., 2, 225-234, doi:10.1021/acsearthspacechem.7b00123, 2018.

Kortelainen, M., Jokiniemi, J., Tiitta, P., Tissari, J., Lamberg, H., Leskinen, J., Grigonyte-Lopez Rodriguez, J., Koponen, H., Antikainen, S., Nuutinen, I., Zimmermann, R., and Sippula, O.: Time-resolved chemical composition of small-scale batch combustion emissions from various wood species. Fuel 233, 224-236, doi:10.1016/j.fuel.2018.06.056, 2018.

Li, R., Palm, B. B., Ortega, A. M., Hlywiak, J., Hu, W., Peng, Z., Day, D. A., Knote, C., Brune, W. H., De Gouw, J. A. and Jimenez, J. L.: Modeling the radical chemistry in an oxidation flow reactor: radical formation and recycling, sensitivities, and the OH exposure estimation equation, The Journal of Physical Chemistry A, 119, 4418-4432, doi:10.1021/jp509534k, 2015.

Nielsen, I. E., Eriksson, A.C., Lindgren R., Martinsson, J., Nyström, R., Nordin, E. Z., Sadiktsis, J., Boman, C., Nojgaard, J. K., and Pagels, J.: Time-resolved analysis of particle emissions from residential biomass combustion – Emissions of refractory black carbon, PAHs and organic tracers. Atm. Environ. 165, 179-190, doi:10.1016/j.atmosenv.2017.06.033, 2017.

Peng, Z., Lee-Taylor, J., Orlando, J. J., Tyndall, G. S. and Jimenez, J. L.: Organic peroxy radical chemistry in oxidation flow reactors and environmental chambers  and their atmospheric relevance, Atmospheric Chem. Phys., 19(2), 813–834, doi:10.5194/acp-19-813-2019, 2019.

Reda, A. A., Czech, H., Schnelle-Kreis, J., Sippula, O., Orasche, J., Weggler, B., Abbaszade, G., Arteaga-Salas, J., Kortelainen, M., Tissari, J., Jokiniemi, J., Streibel, T. and Zimmermann, R.: Analysis of Gas-Phase Carbonyl Compounds in Emissions from Modern Wood Combustion Appliances: Influence of Wood Type and Combustion Appliance, Energy Fuels, 29, 3897-3907, doi:10.1021/ef502877c, 2015.

---

## Author Response (AR2)

We thank the editor for these comments. In the following, we offer point-to-point responses to the remaining questions, with the replies indicated as red text. Line numbers in the responses refer to lines in the revised manuscript without changes visible. The manuscript with changes visible is available after these responses.

**Editor Decision: Publish subject to minor revisions (review by editor)** (11 Apr 2020) by Sergey A. Nizkorodov
Comments to the Author:
Dear authors.
Please address the reviewer remaining comments reproduced here: "I have two remaining comments.

- o First is that the limitations of using the OFR should be acknowledged in the conclusions section of the paper. In particular, in the second to last paragraph the differences in dilution in the real atmosphere versus the OFR and the implications for changes in particle composition need to be acknowledged.

Effect on dilution on partitioning of OA is now noted in the conclusions section (l. 631-632).

- o Second, I think that the pie charts of peroxy radical fate in the OFR included in the response document are valuable and should be included in the SI. This chart allows the reader to fully appreciate how much the RO2+OH reaction is enhanced (non track changes version line 367).

Pie charts showing the estimated $RO_2$ fates in ambient air and in two specific conditions of this study were included in the SI section as Fig. S8 and referenced to in the main text on line 370.

I only have minor suggestions:

- o Line 83: you may want to use classic references to mutagenicity of PACs by Pitts from the 60s, which are also summarized in Chapter 10 of this book: "Finlayson-Pitts, B. J.; Pitts, J. N., Chemistry of the Upper and Lower Atmosphere: Theory, Experiments, and Applications. Academic Press: San Diego, 2000; p 969 pp."

Reference was included on line 83.

- o Figure 4: should all the factions shown add up to 1? Two of the bars (6.8d in S-5% and 4.0d in S-22%) do not seem to, and it is not clear what the base for the normalization is in that case.

Figure 4 (now Fig. 3) was normalized to the total OGC content from the PTR-MS spectra and still included a few ions outside the grouping, which is why the bars of the groups did not add up to 1. As pointed out by the editor, normalization of the bars to the relative share of the identified compound groups makes the figure clearer to readers, and the figure was revised accordingly.

- o Figure numbering in the text needs some work. Figures in the paper are called almost in order: 1, 2, 4, 3, 5, 6, 7, 8, 9, 10, so only one permutation is needed. However, the supporting information figures are completely out of order, appearing in sequence of: S9, S5, S7, S1, S2, S3, S4, S15, S6, S10, S11, S8, S14, S12, S13, S17, S20, S16, S18, S19. The number of supporting information figures make the paper a little overwhelming to read. I would recommend prioritizing them and perhaps removing ones with the lowest level of priority.

Numbering of supplementary tables and figures was revised. In the revised version, figures and tables appear in the order of reference. Four of the original supplementary figures (S2, S4, S12 and S13) were removed, and one figure (Fig. S8) regarding the $RO_2$ fate was added, as requested by the reviewer.

- o Given the size of it, the SI section would strongly benefit from a table of contents.

Table of contents was added to the beginning of SI.

[revised manuscript text omitted]